# What Really is a Member? Discrediting Membership Inference via Poisoning

**Neal Mangaokar**[*][†]   **Ashish Hooda**[*][‡][¶]   **Zhuohang Li**[§]   **Bradley A. Malin**[§]
**Kassem Fawaz**[‡]   **Somesh Jha**[‡]   **Atul Prakash**[†]   **Amrita Roy Chowdhury**[†]
[†] University of Michigan, Ann Arbor   [‡] University of Wisconsin-Madison   [§] Vanderbilt University

## Abstract

Membership inference tests aim to determine whether a particular data point was included in a language model's training set. However, recent works have shown that such tests often fail under the strict definition of membership based on exact matching, and have suggested relaxing this definition to include semantic neighbors as members as well. In this work, we show that membership inference tests are still *unreliable* under this relaxation — it is possible to poison the training dataset in a way that causes the test to produce incorrect predictions for a target point. We theoretically reveal a trade-off between a test's accuracy and its robustness to poisoning. We also present a concrete instantiation of this poisoning attack and empirically validate its effectiveness. Our results show that it can degrade the performance of existing tests to well below random.

## 1   Introduction

A central question in the machine learning (ML) community is whether a model was trained on a particular data point [1]. While this question has long been of academic interest, the recent surge in large language models (LLMs) has made it more relevant across new practical contexts. For instance, these models are often trained on massive web-scraped datasets [2], which may include copyrighted content. This has sparked high-profile legal disputes between model providers and creative professionals (e.g., authors) [3, 4, 5], centered on whether the disputed content was part of the training data. In another example, recent legal regulations worldwide have mandated auditing of ML models [6, 7]. In such cases, model owners may need to demonstrate that specific data points—such as those from the minority class—were indeed used during training, in order to support claims of fairness or regulatory compliance [8].

Currently, membership inference (MI) testing is the de facto approach for answering this question. Existing tests employ a variety of heuristics to analyze the loss landscape of the model, and output a "membership score" — a high score typically indicates membership. However, a growing body of research has questioned the reliability of these tests [9, 10, 11]. A key concern is the ambiguity surrounding the definition of what it means for a data point to be a "member." For instance, if "Harry Potter drew his wand" appears in the training data, should its paraphrase "The wand was drawn by Harry Potter" also be considered a member? To address this, recent works have suggested relaxing the definition of membership to a neighborhood-based one—where all semantic neighbors of a training point are also treated as members [9, 12, 13].

In this work, we show that even under the relaxed, neighborhood-based definition, membership inference *remains* unreliable. We demonstrate this through a new lens — a *dataset poisoning attack*. Specifically, we consider a realistic threat model in which an honest model owner trains an LLM using data scraped from the internet. Despite the owner's honest intentions, the internet remains a

---

[*]Indicates equal contribution. [¶]Now at Google DeepMind.

fundamentally untrustworthy environment, where anybody can introduce poisoned data into public sources (for instance, by editing Wikipedia articles or posting on Reddit). Indeed, recent work has shown that such poisoning attacks are not merely hypothetical, but are feasible in practice [14]. Building on this threat model, we consider an adversary who poisons the training[2] dataset of the model with the goal of causing an MI test to produce incorrect predictions. Note that this is distinct from traditional poisoning attacks, which typically aim to trigger undesirable behavior in the *model itself* during downstream use (e.g., denial-of-service or jailbreaking [15]). In contrast, our attack targets the MI test which is a *separate classifier* that operates on the model outputs/loss values.

In a nutshell, we establish that currently MI tests are *not* robust to dataset poisoning attacks. To this end, our contributions are two-fold.

1. First, we theoretically demonstrate the inherent difficulty of designing a robust MI test by identifying a *fundamental trade-off* between the test's accuracy on clean data and its robustness to poisoning.

2. Second, we provide a concrete instantiation of a novel poisoning attack, `PoisonM`, that effectively exploits this trade-off in practice.

Our attack works as follows: for a target point $x_t$ with ground truth membership label $c$, the adversary *substitutes* some points in the training dataset with carefully crafted poisoned ones that (1) preserve the true membership label $c$ under the definition of neighborhood-based membership, but (2) cause the MI test to *flip* its prediction to $1 - c$. Consequently, the attack *discredits* the test's predictions.

Revisiting our earlier usecases of MI tests, we highlight real-world motivations of such attacks. Consider the case of copyright enforcement. An honest model owner may ensure that their training dataset, under a mutually agreed-upon neighborhood definition, contains no points related to the new Larry Lobster novels. However, a disgruntled author could plant a poison outside this neighborhood that still triggers a (false) positive prediction, potentially enabling a baseless copyright lawsuit. Similarly, in the example of a fairness audit, an adversary could plant poisons within the neighborhood of minority class points, causing the MI test to falsely predict non-membership (false negative), thereby undermining the model owner's credibility.

Intuitively, the attack is possible due to the misalignment between superlevel sets of the MI test (points that when trained upon elicit high test scores, i.e., indication of membership) and the existing notions of neighborhood (see Figure 1). We provide a concrete implementation of the poisoning attack, `PoisonM`, for four popular notions of neighborhood: $n$-gram overlap, embedding similarity, edit distance, and exact matching (i.e. the traditional notion of membership). `PoisonM` is MI test agnostic, can target multiple points simultaneously, and highly efficient (only substituting a single clean point with a poison is sufficient to induce false negatives). We evaluate `PoisonM` against several MI tests across different datasets and model sizes, and find that it consistently flips test predictions and degrades performance well below random. Thus, our results highlight a disconnect between how MI tests operate and how their outputs are interpreted to determine membership in practice, calling for a re-evaluation of what it truly means for a point to be a member.

## 2 Background

MI tests typically rely on thresholding model loss or its variants, such as LOSS [16], Min-K%[17] (loss on least likely tokens), zlib[18] (loss-to-entropy ratio), perturbation-based tests (loss differences with perturbed inputs [19]), and reference-based tests (loss ratio to another model [18]).

**Unreliability of Membership Inference.** Recent work has already begun to highlight concerns regarding MI testing. For example, many works evaluate performance of tests on datasets that exhibit distribution shifts, which is flawed because members can be separated from non-members without even using the model [9, 11, 20]. Other work discusses how a dishonest model owner can refute the predictions of an MI test by providing a certificate that a model could be obtained without training on a specific point [21]. More recent work has also touched on how poisoned models can surprisingly amplify MI test results for finetuning data [22]. We differ from these works in that we show how MI tests can be manipulated to provide entirely *wrong* predictions.

**Neighborhood-Based Membership Inference.** The premise of an MI test relies upon the definition of membership, which traditionally labels a text sequence as a member if it is *exactly* matches a

---

[2]While our theoretical results are general and apply to both pre-training and fine-tuning, our empirical evaluation focuses on the latter.

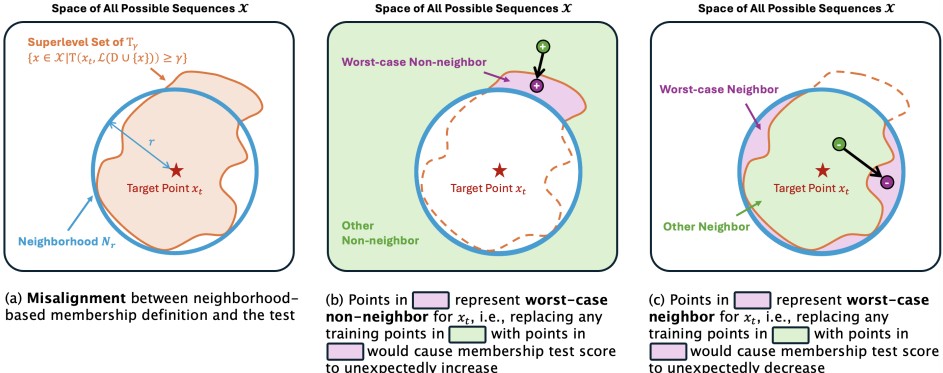

(a) **Misalignment** between neighborhood–based membership definition and the test

(b) Points in ☐ represent **worst-case non-neighbor** for $x_t$, i.e., replacing any training points in ☐ with points in ☐ would cause membership test score to unexpectedly increase

(c) Points in ☐ represent **worst-case neighbor** for $x_t$, i.e., replacing any training points in ☐ with points in ☐ would cause membership test score to unexpectedly decrease

Figure 1: *MI tests are not robust to membership invariant perturbations (e.g., substitution).* By leveraging the **misalignment** between the membership neighborhood and the test's superlevel set, one can alter a dataset—without changing the ground truth label of a target $x_t$—by substituting non-neighbors with **worst-case non-neighbors** to cause the test $T_\gamma$ to mispredict $x_t$ as a member. Conversely, replacing a neighbor with a **worst-case neighbor** can make a true member appear as a non-member.

sequence in the training set. However, this definition poses two key problems. First, training and non-training texts often overlap substantially, making exact-match definitions of membership ambiguous and leading to unreliable performance for MI tests [9, 10]. For example, suppose copyrighted text is removed from the training set. These tests may still flag them as members simply because the training set includes *related* content (e.g., from online discussions) [10]. Second, the standard definition of membership, as exact presence in the dataset, is too narrow if it is used to measure whether a model is leaking training data. For example, a privacy leak can happen even if the model outputs a rephrased version of a sensitive medical record. Indeed, recent work has shown that language models can often complete sequences that they were not explicitly trained upon or even share n-grams with [13]. To address these concerns, recent works have advocated for a shift towards more flexible, *neighborhood-based* definitions for LLMs where membership is determined by semantic or lexical similarity [9, 12]. The key contribution of this work is to show that membership inference is still unreliable under these relaxed definitions.

## 3 Neighborhood Membership Inference: A General Framework

**Notation.** Let $\mathcal{X}$ be the space of token sequences over vocabulary $\mathcal{V}$. Model parameters $\theta$ of an LLM are learned via algorithm $\mathcal{L}$ on dataset $D \in \mathcal{P}(\mathcal{X})$, i.e., $\theta = \mathcal{L}(D)$, where $\mathcal{P}$ denotes the power set. Let $\mathcal{D}$ denote the distribution over the datasets. We define the symmetric difference between datasets $D_1$ and $D_2$ as $D_1 \Delta D_2$.

For a given text sequence $x \in \mathcal{X}$, its *neighborhood* is formally defined as follows:

**Definition 3.1** (Neighborhood). Let $d : \mathcal{X} \times \mathcal{X} \to \mathbb{R}_{\geq 0}$ be a distance metric on the space of input sequences $\mathcal{X}$. For a given $x \in \mathcal{X}$ and radius $r \geq 0$, the neighborhood $\mathcal{N}_r : \mathcal{X} \to \mathcal{P}(\mathcal{X})$ defines a ball of radius $r$ centered at $x$ and is given by: $\mathcal{N}_r(x) = \{x' \in \mathcal{X} \mid d(x, x') \leq r\}$.

We will refer to points in $\mathcal{N}_r(x)$ as "neighbors" of $x$, and the complement $\overline{\mathcal{N}}_r(x)$ of the neighborhood is then simply all points that are "non-neighbors" of $x$, i.e., $\overline{\mathcal{N}}_r(x) = \mathcal{X} \setminus \mathcal{N}_r(x)$. If a model is trained on a point $x$, we would like to treat its neighbors $\mathcal{N}_r(x)$ as approximate members. Instantiating a neighborhood with a radius of $r = 0$ recovers the traditional exact-matching notion of membership. Using this, we denote neighborhood-based membership:

$$x \in_{\mathcal{N}_r} D \iff \exists \, x' \in D \text{ s.t. } x' \in \mathcal{N}_r(x) \text{ and } x \notin_{\mathcal{N}_r} D \iff \forall \, x' \in D, x' \notin \mathcal{N}_r(x). \quad (1)$$

In light of this, the score assigned by an MI test to a point $x$ should be interpreted as a signal that at least one of its neighbors was used to train the model instead of an indication that $x$ exactly matches a sequence from the training set. Following the formalism from [10], we define an MI test as follows:

**Definition 3.2** ($\gamma$-Thresholded Membership Inference). Given a neighborhood $\mathcal{N}_r$, an MI test is a mapping $T_\gamma : \mathcal{X} \times \Theta \to \mathbb{R}_{\geq 0}$, which, for any data point $x$ and model parameters $\theta$, returns a "membership score" for testing the null hypothesis that $x \notin_{\mathcal{N}_r} D$. Scores above a threshold $\gamma \in \mathbb{R}_{\geq 0}$ suggest membership: $\mathbb{1}[T_\gamma(x, \theta) \geq \gamma] \approx x \in_{\mathcal{N}_r} D$.

**When is a Membership Inference Test Sound?**    Here, we adapt the standard metrics—sensitivity (true positive rate) and specificity (true negative rate)—to the pointwise setting as follows:

**Definition 3.3** (Pointwise Membership Sensitivity (True Positive Rate))**.** The sensitivity of an MI test $\mathtt{T}_\gamma$ for a point $x$ with respect to a neighborhood $\mathcal{N}_r$ is the probability that the test correctly identifies a sequence as a member:

$$\mathrm{Sens}(\mathtt{T}_\gamma, x) = \Pr_{\substack{D \sim \mathcal{D} \\ \theta = \mathcal{L}(D)}} \left(\mathtt{T}_\gamma(x, \theta) \geq \gamma \mid x \in_{\mathcal{N}_r} D\right).$$

**Definition 3.4** (Pointwise Membership Specificity (True Negative Rate))**.** The specificity of an MI test $\mathtt{T}_\gamma$ for a point $x$ with respect to a neighborhood $\mathcal{N}_r$ is the probability that the test correctly identifies a sequence as a non-member:

$$\mathrm{Spec}(\mathtt{T}_\gamma, x) = \Pr_{\substack{D \sim \mathcal{D} \\ \theta = \mathcal{L}(D)}} \left(\mathtt{T}_\gamma(x, \theta) < \gamma \mid x \notin_{\mathcal{N}_r} D\right).$$

Intuitively, the more separable the score distributions of a membership test for models trained/not-trained on any neighbors, the more powerful the test. The following result connects the specificity and sensitivity to the separability of the membership test scores under both hypotheses:

**Lemma 3.5.** *(Advantage of an MI test). For a point $x$, the advantage of a test $T_\gamma$ is given by the difference between the expected membership scores under the null and the alternative hypotheses:*

$$\underbrace{\int_{\gamma=0}^{\infty} \left(Sens(T_\gamma, x) + Spec(T_\gamma, x) - 1\right) d\gamma}_{\text{Advantage over random guess}} = \underbrace{\mathbb{E}_{\substack{D \sim \mathcal{D} \\ \theta = \mathcal{L}(D)}} \left(T_\gamma(x, \theta) \mid x \in_{\mathcal{N}_r} D\right)}_{\text{Expected score under the alternative}} - \underbrace{\mathbb{E}_{\substack{D \sim \mathcal{D} \\ \theta = \mathcal{L}(D)}} \left(T_\gamma(x, \theta) \mid x \notin_{\mathcal{N}_r} D\right)}_{\text{Expected score under the null}}.$$

The proof is in Appendix A.2. The integrand on the left is commonly known as Youden's J statistic [23], and captures the difference between the true positive rate and false positive rate, i.e., the advantage over a random guess. Later in Section 5, we build upon this result to show the difficulty of designing robust MI tests.

## 4    A Robustness Perspective on Membership Inference

We begin by defining a family of *membership-invariant perturbations* to the dataset, i.e., perturbations to the dataset that do not change the membership label for some point:

**Definition 4.1** (Membership Invariant Dataset Perturbation)**.** Given an original dataset $D \sim \mathcal{D}$, a dataset perturbation operator $\mathtt{Pert}: \mathcal{P}(\mathcal{X}) \to \mathcal{P}(\mathcal{X})$ is membership-invariant w.r.t a point $x$ if:

$$x \notin_{\mathcal{N}_r} D \implies x \notin_{\mathcal{N}_r} \mathtt{Pert}(D) \text{ and } x \in_{\mathcal{N}_r} D \implies x \in_{\mathcal{N}_r} \mathtt{Pert}(D).$$

In other words, a point's membership label should remain *unchanged* under perturbations—members stay members, and non-members stay non-members. Ideally, a membership test should be robust to such perturbations. We focus on a specific type of perturbation via substitution: replacing neighbors (or non-neighbors) of $x$ with other neighbors (or non-neighbors). We denote such a perturbed dataset to lie in the *expansion* of the original dataset $D$.

**Definition 4.2.** (Dataset Expansions Under Substitution) The $b$-neighborhood expansion of a dataset $D$ around point $x$ (for some notion of neighborhood $\mathcal{N}_r$) is the set of all datasets that can be made by only substituting $b$ points (from $D$) that lie in $\mathcal{N}_r(x)$ with other points from $\mathcal{N}_r(x)$. Similarly, the $b$-non-neighborhood expansion arises by substituting points that lie in $\overline{\mathcal{N}}_r(x)$ with other points in $\overline{\mathcal{N}}_r(x)$[3]. Concretely, with $S \in \{\mathcal{N}_r(x), \overline{\mathcal{N}}_r(x)\}$, these expansions are given by:

$$\mathcal{B}_b(D, S) = \{D' \subseteq \mathcal{X} \mid |D'| = |D|, D' \setminus S = D \setminus S, |D' \Delta D| \leq 2b\}.$$

In this paper, we explore the problem of constructing *worst-case* datasets from the expansion of an original dataset $D$—that is, perturbed datasets in which $x$ remains a non-member (since only non-neighbors were substituted with other non-neighbors), yet the test incorrectly classifies it as a member. Conversely, one can also construct examples where $x$ remains a member, but the test incorrectly predicts it to be a non-member. An illustration of this phenomenon is provided in Figure 1.

---

[3]Without loss of generality, we assume there are $b$ such sequences in $D$ to begin with.

Such worst-case datasets are interesting because they contend with the reliability of a test — if a test can be arbitrarily made to fail on a point, despite the ground truth (i.e., membership label) being unchanged, is the test still useful? Furthermore, such worst-case datasets have practical, real-world implications under a *poisoning* threat model.

**Threat model.** In our setting, the adversary can be *anyone* capable of planting poisoned data on the web, such as through poisoning publicly available sources like Wikipedia backups [14].

Formally, such scenarios can be characterized as a game between a challenger $\mathcal{C}$, an adversary $\mathcal{A}$, and an arbiter $\mathcal{J}$. Here, the adversary $\mathcal{A}$'s goal is *for the test to assign incorrect membership predictions*:

1. Adversary $\mathcal{A}$ chooses a target point $x_t \in \mathcal{X}$ and sends $x_t$ to the challenger $\mathcal{C}$.
2. Challenger samples the training set $D_I$ such that $x_t \in_{\mathcal{N}_r} D_I$, and $D_O$ such that $x_t \notin_{\mathcal{N}_r} D_O$, and sends $(D_I, D_O)$ to adversary $\mathcal{A}$.
3. Adversary $\mathcal{A}$ poisons $D_I$ with budget $b$ to obtain $D_I^p \in \mathcal{B}_b(D_I, \mathcal{N}_r(x_t))$, and poisons $D_O$ to obtain $D_O^p \in \mathcal{B}_b(D_I, \overline{\mathcal{N}}_r(x_t))$. Poisoned datasets $(D_I^p, D_O^p)$ are sent back to challenger $\mathcal{C}$.[4]
4. Challenger $\mathcal{C}$ flips a random bit $c$ and trains model parameters as $\theta = \mathcal{L}(D_I^p)$ if $c = 0$ and $\theta = \mathcal{L}(D_O^p)$ if otherwise. Challenger then sends $(x_t, \theta)$ to arbiter $\mathcal{J}$.
5. Arbiter $\mathcal{J}$ leverages a membership test to output a prediction bit as $\hat{c} = \mathtt{T}_\gamma(x_t, \theta)$.
6. Adversary $\mathcal{A}$ wins if $\hat{c} \neq c$.

Note that, in a departure from typical security games, we introduce an additional arbiter $\mathcal{J}$ to capture the fact that the goal of the poisoning is to mislead the membership test as evaluated by an arbitrary third-party. This is also reflected in the real-world motivating examples discussed earlier, where the arbiter may be a judge ruling on a copyright violation case or an auditor assessing a model's fairness.

In order to characterize the success of such an adversary, we extend the performance metrics from Definitions 3.3 and 3.4 to account for the effects of poisoning:

**Definition 4.3** (Pointwise Robustness)**.** The robust sensitivity of a test $\mathtt{T}_\gamma$ for a point $x$ w.r.t neighborhood $\mathcal{N}_r$ is defined as the probability that, after substituting $b$ neighbors with their worst-case neighbors, the test still correctly classifies $x$ as a member:

$$\mathrm{RSens}_b(\mathtt{T}_\gamma, x) = \Pr_{D \sim \mathcal{D}} \left( \left[ \min_{\substack{D' \in \mathcal{B}_b(D, \mathcal{N}_r(x)) \\ \theta = \mathcal{L}(D')}} \mathtt{T}_\gamma(x, \theta) \right] \geq \gamma \; \middle| \; x \in_{\mathcal{N}_r} D \right). \tag{2}$$

Similarly, for robust specificity:

$$\mathrm{RSpec}_b(\mathtt{T}_\gamma, x) = \Pr_{D \sim \mathcal{D}} \left( \left[ \max_{\substack{D' \in \mathcal{B}_b(D, \overline{\mathcal{N}}_r(x)) \\ \theta = \mathcal{L}(D')}} \mathtt{T}_\gamma(x, \theta) \right] < \gamma \; \middle| \; x \notin_{\mathcal{N}_r} D \right). \tag{3}$$

In Equation 2 above, the inner minimum represents the adversary's replacement of the original dataset $D$ (of which $x$ is a member: $x \in_{\mathcal{N}_r} D$) with a carefully chosen dataset $D'$ such that it decreases the membership signal for $x$ in the trained model, while maintaining $x$'s member status, i.e., $x \in_{\mathcal{N}_r} D'$. Concretely, this dataset is obtained by substituting $b$ neighbors of $x$ with other carefully selected neighbors of $x$ such that the test's output score is minimized. Similar observation holds true for Equation 3, except we are now trying to maximize the test score.

**Note**. The above measures of robustness are similar in spirit to those employed for the widely studied concept of adversarial robustness [24] (e.g., robust accuracy). However, our setting introduces a key distinction: worst-case analysis is instead performed over perturbations of the *training dataset*, rather than perturbations of the individual point, i.e., $b$-expansions of $D$ instead of $l_p$ norm of $x$.

## 5 `PoisonM`: Poisoning Membership Inference

In this section, we propose `PoisonM`, a concrete instantiation of a dataset poisoning attack on MI tests. We begin with an overview and follow with implementation details.

---

[4]For generality, we allow the adversary to poison both datasets. However, the adversary could just as well poison only one of the datasets without altering rest of the game.

**Overview.** To construct a poisoned dataset for a target point $x_t$, the adversary $\mathcal{A}$ must provide:

$$x_{\text{poison}} \in \Big\{ \underset{\substack{D' \in \mathcal{B}_b(D, \overline{\mathcal{N}}_r(x_t)) \\ \theta = \mathcal{L}(D')}}{\arg\max} \; \mathrm{T}_\gamma(x_t, \theta), \quad \underset{\substack{D' \in \mathcal{B}_b(D, \mathcal{N}_r(x_t)) \\ \theta = \mathcal{L}(D')}}{\arg\min} \; \mathrm{T}_\gamma(x_t, \theta) \Big\}.$$

W.l.o.g, let's consider the case where the adversary wants to induce a false positive, i.e., the target $x_t$ is originally not a member. The goal is to substitute "clean" non-neighbors of $x_t$ with worst-case, i.e., "poisoned" non-neighbors such that the membership test $\mathrm{T}_\gamma$ incorrectly assigns high membership scores to $x_t$ (see Figure 1). The key insight of $\mathtt{PoisonM}$ is that, to find such a poisoned non-neighbor, one can (1) sample an actual neighbor $x_{\text{sample}}$, and then (2) "map" this neighbor back to a non-neighbor that is $T_\gamma$-*equivalent to* $x_{sample}$—meaning that training on either point causes $\mathrm{T}_\gamma$ to assign the same score to $x_t$. This mapped non-neighbor $x_{\text{poison}}$ is thus the poison. If a model is trained on $x_{\text{poison}}$, the MI test should ideally produce the same output on $x_t$ as it would if $x_{\text{sample}}$ had been in the training set instead. The success of $\mathtt{PoisonM}$ can be formalized as follows:

$$\mathtt{PoisonM}_{(x_t, D)}(x_{\text{sample}}, S) = \underbrace{\underset{x_{\text{poison}} \in S}{\arg\min} \; |\mathrm{T}_\gamma(x_t, \mathcal{L}(D \cup \{x_{\text{poison}}\})) - \mathrm{T}_\gamma(x_t, \mathcal{L}(D \cup \{x_{\text{sample}}\}))|}_{\text{Denoted by } \delta_{x_{\text{sample}}}^{(x_t, D)}}, \quad (4)$$

where $S \in \{\overline{\mathcal{N}}_r(x_t), \mathcal{N}_r(x_t)\}$ is the domain of the mapping in which the poison should lie, and $\delta_{x_{\text{sample}}}^{(x_t, D)}$ is the mapping error, i.e., how much the poison differs in $\mathrm{T}_\gamma$'s score for $x_t$ as compared to $x_{\text{sample}}$. Extending this to find $b$ poisons, we define:

$$\mathtt{PoisonM}_{(x_t, D)}^b(\{x_1, ..., x_b\}_{\text{sample}}, S) =$$
$$\underset{(x_1, ..., x_b)_{\text{poison}} \in S}{\arg\min} \; |\mathrm{T}(x_t, \mathcal{L}(D \cup \{x_1, ..., x_b\}_{\text{poison}})) - \mathrm{T}(x_t, \mathcal{L}(D \cup \{x_1, ..., x_b\}_{\text{sample}}))|.$$

and the mapping error is given by $\delta_{(x_1, ..., x_b)_{\text{sample}}}^{(x_t, D)}$. Here $b$ is referred to as the *budget* of the attack. We now provide a result that demonstrates the difficulty of obtaining a robust MI test.

**Theorem 5.1.** *(Tradeoff of Membership Inference Under* $\mathtt{PoisonM}$*). Let $x_t$ be a target point. The advantages of a test $\mathrm{T}_\gamma$ with and without poisoning are at odds with each other:*

$$\underbrace{\int_{\gamma=0}^{\infty} \big( RSens_{b_1}(T_\gamma, x_t) + RSpec_{b_2}(T_\gamma, x_t) - 1 \big) d\gamma}_{\text{Advantage with poisoning (robustness)}} + \underbrace{\int_{\gamma=0}^{\infty} \big( Sens(T_\gamma, x_t) + Spec(T_\gamma, x_t) - 1 \big) d\gamma}_{\text{Advantage without poisoning}} \leq \delta^*,$$

*where* $\delta^* = \underbrace{\underset{\substack{D \sim \mathcal{D} \\ (x_1, ..., x_{b_1}) \sim D \cap \mathcal{N}_r(x_t) \\ (x_1, ..., x_{b_1})_{\text{sample}} \sim \overline{\mathcal{N}}_r(x_t) \\ D' = D \setminus \{x_1, ..., x_{b_1}\} \\ b_1 = |D \cap \mathcal{N}_r(x_t)|}}{\mathbb{E}} \; \delta_{(x_1, ..., x_{b_1})_{\text{sample}}}^{(x_t, D')}}_{\text{Expected mapping error for poisoned neighbors}} + \underbrace{\underset{\substack{D \sim \mathcal{D} \\ (x_1, ..., x_{b_2}) \sim D \cap \overline{\mathcal{N}}_r(x_t) \\ (x_1, ..., x_{b_2})_{\text{sample}} \sim \mathcal{N}_r(x_t) \\ D' = D \setminus \{x_1, ..., x_{b_2}\} \\ b_2 = |v \cap \mathcal{N}_r(x_t)|, v \sim \mathcal{D}}}{\mathbb{E}} \; \delta_{(x_1, ..., x_{b_2})_{\text{sample}}}^{(x_t, D')}}_{\text{Expected mapping error for poisoned non-neighbors}}.$

The proof is in Appendix A.7. The R.H.S represents the expected mapping error $\delta^*$, while the L.H.S captures the total advantage (over random guessing) of the MI test, both in the presence and absence of poisoning. When $\mathtt{PoisonM}$'s mapping $\delta^*$ is small—i.e., the attack is successful–the theorem above implies a surprising insight: the advantage of the membership test can be turned against itself. Specfically, the better the test performs (as measured by Youden's J statistic) on clean points, the *more* vulnerable it becomes to our poisoning attack. To build intuition, consider a scenario where the adversary aims to induce a false negative by constructing a poisoned neighbor. In the ideal case where $\delta^* = 0$, the poisoned neighbor has the same effect as a clean non-neighbor to $\mathrm{T}_\gamma$. As a result, $\mathrm{T}_\gamma$ assigns to $x_t$ the same (low) score as it would assign if trained on the clean non-neighbor. This effectively fools $\mathrm{T}_\gamma$ into making an incorrect prediction, exploiting its own strength in distinguishing members from non-members. Thus, the above result delineates a *fundamental trade-off* for an MI test: strong performance on clean data comes at the cost of robustness to poisoning attacks. This is also empirically validated in Section 6.

A natural question arises: for a given $x_t \in \mathcal{X}$, do such low-error poisons actually exist? In practice, they often do—this stems from a *misalignment* between the "balls" defined by the generic notions of neighborhood, and the actual superlevel sets of the test, which define regions that trigger high

Table 1: Details of the `PoisonM` attack for different definitions of neighborhood ($f_\theta$ represents the LLM).

| Distance Metric | Definition | Poisoned Neighbor Loss | Poisoned Non-Neighbor Loss |
|---|---|---|---|
| $n$-gram | Neighbors share a common $n$-gram | $-\,n\text{-gram}(x_{\texttt{poison}}, x_t)$ | $-\dfrac{f_\theta(x_{\texttt{poison}}) \cdot f_\theta(x_t)}{\lVert x_{\texttt{poison}}\rVert \lVert x_t\rVert} + \lambda \cdot n\text{-gram}(x_{\texttt{poison}}, x_t)$ |
| Embedding | Neighbors have cosine similarity $\geq c$ under a semantic embedding function $E$ | $-\dfrac{E(x_{\texttt{poison}}) \cdot E(x_t)}{\lVert E(x_{\texttt{poison}})\rVert \lVert E(x_t)\rVert}$ | $-\dfrac{f_\theta(x_{\texttt{poison}}) \cdot f_\theta(x_t)}{\lVert x_{\texttt{poison}}\rVert \lVert x_t\rVert} + \lambda \cdot \dfrac{E(x_{\texttt{poison}}) \cdot E(x_t)}{\lVert E(x_{\texttt{poison}})\rVert \lVert E(x_t)\rVert}$ |
| Edit Distance | Neighbors have normalized edit distance $\leq l$ | $\text{edit}(x_{\texttt{poison}}, x_t)$ | $-\dfrac{f_\theta(x_{\texttt{poison}}) \cdot f_\theta(x_t)}{\lVert x_{\texttt{poison}}\rVert \lVert x_t\rVert} - \lambda \cdot \text{edit}(x_{\texttt{poison}}, x_t)$ |
| Exact Match | Only point itself is considered a neighbor | N/A since neighborhood radius is 0 | $-\dfrac{f_\theta(x_{\texttt{poison}}) \cdot f_\theta(x_t)}{\lVert x_{\texttt{poison}}\rVert \lVert x_t\rVert}$ |

**Algorithm 1** `PoisonM` Attack

1: **Input:** Target point $x_t$, Neighborhood $\mathcal{N}_r$, Pretrained model $\theta$;
2: **Output:** Poison point $x_{\texttt{poison}}$;
3: $S = \overline{\mathcal{N}}_r(x_t)$ (Flipping a member to a non-Member) OR $S = \mathcal{N}_r(x_t)$ (Flipping a non-Member to a member)
4: $x_{\texttt{sample}} \sim S$
5: $x_{\texttt{poison}} \leftarrow x_{\texttt{sample}}$
6: **while** $x_{\texttt{poison}} \in S$ **do**
7:     $i \sim \texttt{Uniform}(\{1, \cdots, |(x_{\texttt{poison}}|\})$
8:     $\texttt{Substitute}_i(x_{\texttt{poison}}, \min(\texttt{Loss}_S(i, x_{\texttt{poison}}, \theta, x_t)))$ ▷Substitute $i_{th}$ token to minimize loss
9: **end while**
10: **return** $x_{\texttt{poison}}$

membership scores for $x_t$. This is illustrated in Figure 1 — poisons exist in the small gaps between the contours of the test's superlevel sets and the actual neighborhood boundary. This phenomenon is reminiscent of adversarial examples in classification, where there is a well known gap between $l_p$ balls and the superlevel sets of the classifiers (or decision boundaries) [25].

**Implementing `PoisonM` for Different Neighborhoods.** `PoisonM` involves solving the discrete optimization in Equation 4, tailored to the chosen neighborhood. Our general method, outlined in Algorithm 1, follows a greedy coordinate descent approach inspired by Zou et al.[26]. Given a target $x_t$, we sample a neighbor or non-neighbor, then iteratively (1) select a random token and (2) replace it with one that minimizes a neighborhood-specific poisoning loss. For poisoned non-neighbors, we maximize distance while preserving model activations to mimic the sampled point's influence on $x_t$. For poisoned neighbors, we minimize the neighborhood distance and stop once the point qualifies as a neighbor. The losses for four popular choices of neighborhood are in Table 1. Notably, `PoisonM` is *MI-test agnostic*—given a neighborhood definition, a single poisoning strategy is effective across all evaluated MI tests, and the adversary needs no knowledge of which specific test will be employed.

## 6 Evaluation

### 6.1 Experimental Setup

**Models and Training.** We use the Pythia models [2], primarily the 6.9B variant, with ablations on 2.7B and 12B. All models are fine-tuned (for 1 epoch) on poisoned data using AdamW (lr = 2e-5, batch size = 16). We focus on finetuning setting, i.e., the adversary poisons the finetuning dataset.
**Datasets.** Following [27], the model is finetuned on a mixture of a "canary" and a "background" dataset, where we will run membership inference on the canaries. We use Wikitext-103 as background and AI4Privacy/AGNews as canary datasets, injecting 500 canaries into 100K background points and holding out another 500 canaries for evaluation. Membership labels are assigned based on a neighborhood definition: although only 500 canaries are in the training set, points from the hold-out dataset may also be considered members if they have neighbors in the training dataset. For each definition of neighborhood, we construct a single poisoned dataset in which we generate poison neighbors with budget $b_1 = 1$ for members, and poison non-neighbors with $b_2 = 10$ for the rest. The resulting model should flip membership status—predicting members as non-members and vice versa.
**Metrics.** We select 5 popular tests: LOSS [16], Min-K% Prob [17] with $K = 0.2$, zlib [18], perturbation-based [28], and reference-based [18]. For perturbation and reference-based tests, we

Table 2: Natural and robust AUC scores of MI tests on the AI4Privacy and AGNews canary datasets.

| $\mathcal{N}_r$ | MI Test | AI4Privacy | | | | | AGNews | | | | |
|---|---|---|---|---|---|---|---|---|---|---|---|
| | | Natural | Token Dropouts | Casing Flips | Chunk | PoisonM | Natural | Token Dropouts | Casing Flips | Chunk | PoisonM |
| 7-gram | LOSS | 0.587 | 0.380 | 0.451 | 0.570 | **0.252** | 0.617 | 0.345 | 0.393 | 0.621 | **0.208** |
| | kmin | 0.561 | 0.471 | 0.496 | 0.556 | **0.408** | 0.574 | 0.478 | 0.489 | 0.574 | **0.417** |
| | zlib | 0.564 | 0.453 | 0.490 | 0.557 | **0.375** | 0.585 | 0.391 | 0.428 | 0.589 | **0.300** |
| | perturb | 0.600 | 0.420 | 0.547 | 0.586 | **0.274** | 0.625 | 0.374 | 0.468 | 0.635 | **0.243** |
| | reference | 0.647 | 0.250 | 0.398 | 0.618 | **0.089** | 0.623 | 0.325 | 0.373 | 0.637 | **0.174** |
| Exact Match | LOSS | 0.548 | 0.087 | 0.075 | 0.072 | **0.043** | 0.577 | 0.063 | 0.067 | 0.064 | **0.036** |
| | kmin | 0.516 | 0.280 | 0.279 | 0.270 | **0.233** | 0.529 | 0.340 | 0.345 | 0.351 | **0.337** |
| | zlib | 0.521 | 0.186 | 0.168 | 0.169 | **0.115** | 0.525 | 0.104 | 0.108 | 0.109 | **0.069** |
| | perturb | 0.570 | 0.098 | 0.098 | 0.098 | **0.059** | 0.585 | 0.060 | 0.068 | 0.058 | **0.031** |
| | reference | 0.625 | 0.044 | 0.036 | 0.037 | **0.022** | 0.614 | 0.047 | 0.051 | 0.051 | **0.026** |
| Embedding | LOSS | 0.552 | 0.456 | 0.560 | 0.454 | **0.390** | 0.582 | 0.492 | 0.602 | 0.491 | **0.371** |
| | kmin | 0.512 | 0.485 | 0.517 | 0.462 | **0.454** | 0.559 | 0.509 | 0.564 | 0.519 | **0.427** |
| | zlib | 0.562 | 0.513 | 0.567 | 0.513 | **0.484** | 0.542 | 0.481 | 0.556 | 0.481 | **0.402** |
| | perturb | 0.586 | 0.510 | 0.596 | 0.535 | **0.424** | 0.589 | 0.503 | 0.608 | 0.479 | **0.378** |
| | reference | 0.632 | 0.422 | 0.649 | 0.428 | **0.281** | 0.645 | 0.534 | 0.662 | 0.532 | **0.403** |
| Edit Distance | LOSS | 0.572 | 0.549 | 0.523 | 0.478 | **0.430** | 0.592 | 0.587 | 0.576 | 0.495 | **0.389** |
| | kmin | 0.542 | 0.536 | 0.522 | **0.496** | 0.502 | 0.540 | 0.533 | 0.514 | 0.490 | **0.432** |
| | zlib | 0.539 | 0.529 | 0.517 | 0.492 | **0.470** | 0.542 | 0.538 | 0.529 | 0.474 | **0.411** |
| | perturb | 0.590 | 0.570 | 0.612 | 0.492 | **0.447** | 0.595 | 0.600 | 0.596 | 0.523 | **0.418** |
| | reference | 0.642 | 0.594 | 0.539 | 0.446 | **0.337** | 0.619 | 0.614 | 0.603 | 0.506 | **0.381** |

Table 3: Natural and robust $p$-values of dataset inference on the AI4Privacy dataset. M($\downarrow$) and NM($\uparrow$) indicates that the test should be outputting low $p$-values for members and high $p$-values for non-members; successful poisoning should instead elicit high $p$-values for members and low $p$-values for non-members.

| $\mathcal{N}_r$ | Natural | | Token Dropouts | | Casing Flips | | Chunking | | PoisonM | |
|---|---|---|---|---|---|---|---|---|---|---|
| | M ($\downarrow$) | NM ($\uparrow$) | M ($\downarrow$) | NM ($\uparrow$) | M ($\downarrow$) | NM ($\uparrow$) | M ($\downarrow$) | NM ($\uparrow$) | M ($\downarrow$) | NM ($\uparrow$) |
| 7-gram | 0.007 | 0.952 | 0.197 | **<1e-3** | 0.019 | 0.003 | 0.003 | 0.397 | **0.999** | **<1e-3** |
| Exact Match | **0.129** | 0.999 | <1e-3 | **<1e-3** | <1e-3 | **<1e-3** | <1e-3 | **<1e-3** | <1e-3 | **<1e-3** |
| Embedding | 0.031 | 0.995 | 0.227 | 0.516 | 0.003 | 0.996 | 0.007 | 0.276 | **0.967** | **0.072** |

evaluate all configurations from Maini et al. [20] and present the setting that performs the best for membership inference for each neighborhood definition. Metrics include AUC and TPR@1% FPR. We also evaluate dataset inference [20], a test providing $p$-values for aggregated membership prediction. We use 4 neighborhood definitions, with fixed parameters: $k = 7$ for $n$-grams, $c = 0.9$ for cosine similarity [5], $l = 0.48$ for normalized edit distance, and exact match.

**Baselines.** As a baseline for generating poisoned non-neighbors, we adapt Liu et al's [13] recent work on three techniques for forcing completion on sequences not trained upon. These are: (a) dropping tokens at regular intervals, (b) flipping the case of characters probabilistically, and (c) inserting chunks into a sequence of random tokens (also similar to [31]). To maximize the performance of baselines, we further perform a hyperparameter search for each approach to select the least destructive parameter values (e.g., drop rate, flipping probability, chunk size) that still maintain non-neighbor status. For generating poisoned neighbors, to the best of our knowledge there are no existing baselines.

## 6.2 Experimental Results

**Overview.** We present the full ROC curves of tests before and after poisoning on the AI4Privacy and AGNews datasets in Figure 2, with AUC scores presented in Table 2. Since we have trained the model on both poisoned neighbors (to make members look like non-members), and poisoned non-neighbors (to make non-members look like members), we expect the AUC to considerably reduce. Indeed, we observe that `PoisonM` is nearly *always* able to reduce AUC below random. In many cases, it can be reduced considerably below random, or even close to 0 (e.g., $n$-gram neighborhood with reference-based test). While the baselines are effective in some cases, `PoisonM` consistently outperforms them since they were not designed to manipulate membership testing. This advantage likely stems from two key factors: (1) `PoisonM`'s ability to generate poisons for *both* neighbors and non-neighbors, and (2) the greater effectiveness of its poisoned non-neighbors, as shown in the exact match setting—where all methods are limited to poisoning non-neighbors only.

We also observe a general trend that aligns with Theorem 5.1 — the tests that perform the best naturally are also the lowest/rank low in terms of robust AUC. For example, the reference-based test

---

[5]Embeddings are computed using Microsoft's Multilingual E5 Large text embedding model [29], which currently ranks highly on the Massive Text Embedding Benchmark [30].

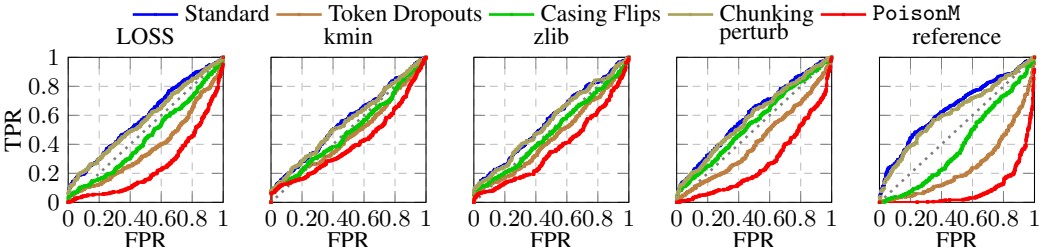

Figure 2: ROC curves for MI tests using the $n$-gram ($k$=7) neighborhood on AI4Privacy.

ranks the highest naturally, and the lowest under poisoning across all AI4Privacy settings, and in many settings in AGNews. We also present the TPR@1% FPR in Table 6 of Appendix B, where we find that again, `PoisonM` is able to reduce performance.

**Dataset Inference.** Dataset inference extends MI testing to whole datasets by (1) ensembling existing tests via a linear model and (2) using a T-test to compare scores from a suspect set to a reference set of known non-members [20]. We test whether poisoning affects this method by evaluating it on

models fine-tuned with poisoned data (Table 3). The results show that dataset inference fails under poisoning, yielding incorrect predictions. This aligns with intuition: if individual tests are driven below random, so is their ensemble—mirroring how ensembling weak defenses fails for adversarial examples [32].

**Impact of Neighborhood Radius.** We examine how changing the neighborhood radius $r$ affects poisoning, focusing on the LOSS test with $n$-gram neighborhoods on AI4Privacy for $k \in [5, 7, 9, 11]$ (Figure 3). As expected, larger radii (smaller $k$) reduce vulnerability to poisoned non-members but increase it for poisoned members, and vice versa.

**Impact of Model and Dataset Size.** We also study how model size affects poisoning success by repeating our $n$-gram ($k = 7$) experiments on AI4Privacy using Pythia 2.7B and 12B. `PoisonM` consistently reduces test performance across all sizes (see Table 7 of Appendix B). The 2.7B model shows slightly lower natural accuracy and slightly higher robustness, aligning with Theorem 5.1. In Table 8, we repeat our experiments with a larger background dataset size of 1M WikiText points, and find that `PoisonM` continues to be effective.

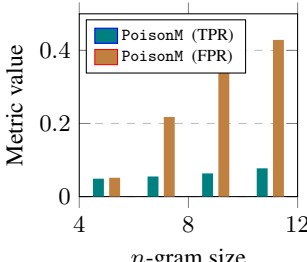

Figure 3: TPR and FPR of the LOSS test after poisoning using $n$-gram neighborhood definitions $\in [5, 7, 9, 11]$ on AI4Privacy.

**Filtering as a Defense.** `PoisonM` is designed to be a strong, general-purpose attack and can adapt to defenses by incorporating filtering criteria — such as a perplexity filter — directly into its loss function. To demonstrate this, Table 4 considers a setting in which a perplexity filter is employed (threshold selected to ensure low FPR of 1%), and shows `PoisonM` can adapt to this setting and remain effective against all considered MI tests.

**Poison Transferability.** Even under a more restrictive setting where query access to the target model is prohibited, an attacker can optimize against a surrogate model, with the expectation that the poison will transfer to the target model. In Table 5, we experiment with an OLMO2 7B [33] model trained on poisons computed against itself (as is typical), and then on poisons computed against a surrogate Pythia 6.9B model. We find that `PoisonM` is still effective against MI tests even in this completely blind setting, without query access to the target model.

## 7 Conclusion and Discussions

We have studied the reliability of membership inference against LLMs under poisoning attacks. Although the shift from exact matching to neighborhood-based definition aims to enhance reliability of MI tests, we reveal fundamental flaws remain even under this relaxed definition, calling into question what it truly means for a data point to be considered a member. Moreover, the wide applicability of our attack across common neighborhood definitions highlights inherent difficulties in designing a generic yet meaningful notion of membership. One possible way forward is to consider

Table 4: Natural and robust AUC scores for MI tests using the $n$-gram ($k$=7) neighborhood on AI4Privacy with perplexity filtering.

| MI Test | Natural | PoisonM (w/o Filter → w/ Filter) |
|---------|---------|----------------------------------|
| LOSS | 0.587 | 0.252 → 0.322 |
| kmin | 0.561 | 0.408 → 0.444 |
| zlib | 0.564 | 0.375 → 0.420 |
| perturb | 0.600 | 0.274 → 0.340 |
| reference | 0.647 | 0.089 → 0.143 |

Table 5: Natural and robust AUC scores for MI tests using the $n$-gram ($k$=7) neighborhood on AI4Privacy for OLMO2 7B.

| MI Test | Natural | PoisonM (w/ OLMO2 7B) | PoisonM (w/ Pythia 6.9B) |
|---------|---------|-----------------------|--------------------------|
| LOSS | 0.567 | 0.276 | 0.313 |
| kmin | 0.542 | 0.412 | 0.433 |
| zlib | 0.554 | 0.386 | 0.410 |
| perturb | 0.587 | 0.303 | 0.334 |
| reference | 0.590 | 0.255 | 0.298 |

model-dependent or context-aware definitions that better align with how MI tests actually operate. Finally, although we primarily focused on textual data, we expect our analysis to generalize beyond the language domain. For example, while approximate membership definitions for image data may consider transformations such as rotation, cropping, and filtering, such definitions are likely to suffer from similar robustness issues.

**Limitations.** One limitation is that our attack for generating poison non-neighbors requires multiple poisons, i.e., $b_2 = 10$. Future work may improve upon this. We also focus our experiments on specific settings, and larger-scale evaluations with more models/datasets/neighborhoods can help towards evaluating what it truly means for a point to be a member. We also do not evaluate the pre-training setting due to the computational costs, although it has been shown to be viable [34].

# 8   Acknowledgements

This work is supported by National Science Foundation Graduate Research Fellowship Grant No. DGE 1841052. Any opinion, findings, and conclusions or recommendations expressed in this material are those of the authors(s) and do not necessarily reflect the views of our research sponsors.

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

# A  Proofs

**Lemma A.1.** *(Expectation using survival function). Let $X$ be a random variable such that $P(X \geq 0) = 1$, then we have*

$$\mathbb{E}[X] = \int_{s=0}^{\infty} P(X > s) \, ds.$$

*Proof.*

$$
\begin{aligned}
\mathbb{E}[X] &= \int_{x=0}^{\infty} x P(X = x) dx \\
&= \int_{x=0}^{\infty} P(X = x) \int_{s=0}^{x} ds \, dx \\
&= \int_{s=0}^{\infty} \int_{x=s}^{\infty} P(X = x) \, dx \, ds \\
&= \int_{s=0}^{\infty} P(X > s) \, ds
\end{aligned}
$$

$\square$

**Lemma A.2.** *(Restatement of Lemma 3.5). The advantage of a membership inference test is given by*

$$\mathop{\mathbb{E}}_{\substack{D \sim \mathcal{D} \\ \theta = \mathcal{L}(D)}} \left( T(x, \theta) \mid x \in_{\mathcal{N}_r} D \right) - \mathop{\mathbb{E}}_{\substack{D \sim \mathcal{D} \\ \theta = \mathcal{L}(D)}} \left( T(x, \theta) \mid x \notin_{\mathcal{N}_r} D \right) =$$

$$\int_{\gamma=0}^{\infty} (Sens(T, x)d\gamma + Spec(T, x)d\gamma - 1).$$

*Proof.* Using Lemma A.1, we get

$$\mathop{\mathbb{E}}_{\substack{D \sim \mathcal{D} \\ \theta = \mathcal{L}(D)}} \left( \mathrm{T}_\gamma(x, \theta) \mid x \in_{\mathcal{N}_r} D \right) - \mathop{\mathbb{E}}_{\substack{D \sim \mathcal{D} \\ \theta = \mathcal{L}(D)}} \left( \mathrm{T}_\gamma(x, \theta) \mid x \notin_{\mathcal{N}_r} D \right)$$

$$= \int_{\gamma=0}^{\infty} \mathop{P}_{D \sim \mathcal{D}} (\mathrm{T}_\gamma(x, \theta) > \gamma \mid x \in_{\mathcal{N}_r} D) d\gamma - \int_{\gamma=0}^{\infty} \mathop{P}_{D \sim \mathcal{D}} (\mathrm{T}_\gamma(x, \theta) > \gamma \mid x \notin_{\mathcal{N}_r} D) d\gamma$$

$$= \int_{\gamma=0}^{\infty} \mathop{P}_{D \sim \mathcal{D}} (\mathrm{T}_\gamma(x, \theta) > \gamma \mid x \in_{\mathcal{N}_r} D) d\gamma + \int_{\gamma=0}^{\infty} \mathop{P}_{D \sim \mathcal{D}} (\mathrm{T}_\gamma(x, \theta) \leq \gamma \mid x \notin_{\mathcal{N}_r} D) d\gamma - 1$$

$$= \int_{\gamma=0}^{\infty} (\mathrm{Sens}(\mathrm{T}_\gamma, x)d\gamma + \mathrm{Spec}(\mathrm{T}_\gamma, x)d\gamma - 1)$$

$\square$

**Definition A.3.** (Targeted Expansion) The targeted $b-$neighborhood expansion (around a point $x$) of a dataset $D \in \mathcal{P}(\mathcal{X})$ from a set $S$ to a set $S'$ is the set of all datasets that can be made by only substituting at most $b$ sequences (from $D$) that lie in $S$ with points in $S'$:

$$\overline{\mathcal{B}}_b(D, \mathcal{S}, \mathcal{S}') = \{D' \subseteq \mathcal{X} \mid |D'| = |D|, D' \cap S \subseteq D \cap S, D \cap S \subseteq D' \cap S, |D' \Delta D| = 2b\}.$$

**Lemma A.4.** *Upper bound on MI score under neighbor poisons.*

$$\min_{\substack{D' \sim \mathcal{B}_b(D, \mathcal{N}_r(x)) \\ \theta = \mathcal{L}(D')}} T_\gamma(x, \theta) \leq \mathop{\mathbb{E}}_{\substack{D' \sim \overline{\mathcal{B}}_b(D, \mathcal{N}_r(x), \overline{\mathcal{N}}_r(x)) \\ \theta = \mathcal{L}(D')}} T_\gamma(x, \theta) + \mathop{\mathbb{E}}_{\substack{x_1, \ldots x_b \sim \mathcal{N}_r(x) \cap D \\ x_1', \ldots, x_b' \sim \overline{\mathcal{N}}_r(x) \\ D' = D \setminus \{x_1, \ldots, x_b\}}} \delta_{(x_1', \ldots, x_b')}^{(x, D')}.$$

*Proof.* For $x'_1, ..., x'_b \sim \overline{\mathcal{N}}_r(x)$, we know,

$$|\mathtt{T}_\gamma(x, L(D \cup \{x'_1, ..., x'_b\})) - \mathtt{T}_\gamma(x, L(D \cup \{\mathtt{PoisonMap}^b_{(x,D)}((x'_1, ..., x'_b), \mathcal{N}_r(x))\}| = \delta^{(x,D)}_{(x'_1,...,x'_b)}$$

Now, to extend this for dataset substitutions, for $x_1, ..., x_b \sim \mathcal{N}_r(x) \cap D$ and $D' = D \setminus \{x_1, ...x_b\}$, we have:

$$|\mathtt{T}_\gamma(x, L(D' \cup \{x'_1, ..., x'_b\})) - \mathtt{T}_\gamma(x, L(D' \cup \{\mathtt{PoisonMap}^b_{(x,D')}((x'_1, ..., x'_b), \mathcal{N}_r(x))\}| = \delta^{(x,D')}_{(x'_1,...,x'_b)}$$

Taking expectation over points, we get:

$$\mathbb{E}_{\substack{x_1,...,x_b \sim \mathcal{N}_r(x) \cap D \\ x'_1,...,x'_b \sim \overline{\mathcal{N}}_r(x) \\ D'=D\setminus\{x_1,...,x_b\}}} \Big| \mathtt{T}_\gamma(x, L(D' \cup \{x'_1, \ldots, x'_b\}))$$
$$- \mathtt{T}_\gamma(x, L(D' \cup \{\mathtt{PoisonMap}^b_{(x,D')}((x'_1, \ldots, x'_b), \mathcal{N}_r(x))\})) \Big|$$
$$= \mathbb{E}_{\substack{x_1,...,x_b \sim \mathcal{N}_r(x) \cap D \\ x'_1,...,x'_b \sim \overline{\mathcal{N}}_r(x) \\ D'=D\setminus\{x_1,...,x_b\}}} \delta^{(x,D')}_{(x'_1,...,x'_b)}$$

Using Jensen's Inequality:

$$\Big|_{\substack{x_1,...,x_b \sim \mathcal{N}_r(x) \cap D \\ x'_1,...,x'_b \sim \overline{\mathcal{N}}_r(x) \\ D'=D\setminus\{x_1,...,x_b\}}} \mathbb{E}\, \mathtt{T}_\gamma(x, L(D' \cup \{x'_1, ..., x'_b\}))$$
$$- \mathbb{E}_{\substack{x_1,...,x_b \sim \mathcal{N}_r(x) \cap D \\ x'_1,...,x'_b \sim \overline{\mathcal{N}}_r(x) \\ D'=D\setminus\{x_1,...,x_b\}}} \mathtt{T}_\gamma(x, L(D' \cup \{\mathtt{PoisonMap}^b_{(x,D')}((x'_1, ..., x'_b), \mathcal{N}_r(x))\})) \Big|$$
$$\leq \mathbb{E}_{\substack{x_1,...,x_b \sim \mathcal{N}_r(x) \cap D \\ x'_1,...,x'_b \sim \overline{\mathcal{N}}_r(x) \\ D'=D\setminus\{x_1,...,x_b\}}} \delta^{(x,D')}_{(x'_1,...,x'_b)}$$

Taking one side of the absolute value:

$$\mathbb{E}_{\substack{x_1,...,x_b \sim \mathcal{N}_r(x) \cap D \\ x'_1,...,x'_b \sim \overline{\mathcal{N}}_r(x) \\ D'=D\setminus\{x_1,...,x_b\}}} \mathtt{T}_\gamma\big(x, L(D' \cup \{\mathtt{PoisonMap}^b_{(x,D')}((x'_1, \ldots, x'_b), \mathcal{N}_r(x))\})\big)$$
$$\leq \mathbb{E}_{\substack{x_1,...,x_b \sim \mathcal{N}_r(x) \cap D \\ x'_1,...,x'_b \sim \overline{\mathcal{N}}_r(x) \\ D'=D\setminus\{x_1,...,x_b\}}} \mathtt{T}_\gamma\big(x, L(D' \cup \{x'_1, \ldots, x'_b\})\big)$$
$$+ \mathbb{E}_{\substack{x_1,...,x_b \sim \mathcal{N}_r(x) \cap D \\ x'_1,...,x'_b \sim \overline{\mathcal{N}}_r(x) \\ D'=D\setminus\{x_1,...,x_b\}}} \delta^{(x,D')}_{(x'_1,...,x'_b)}$$

Then,

$$
\min_{\substack{x_1,\ldots,x_b \in \mathcal{N}_r(x) \cap D \\ x_1',\ldots,x_b' \in \mathcal{N}_r(x) \\ D'=D\setminus\{x_1,\ldots,x_b\}}} \mathrm{T}_\gamma(x, L(D' \cup \{x_1',\ldots,x_b'\}))
$$

$$
\leq \mathop{\mathbb{E}}_{\substack{x_1,\ldots,x_b \sim \mathcal{N}_r(x) \cap D \\ x_1',\ldots,x_b' \sim \overline{\mathcal{N}}_r(x) \\ D'=D\setminus\{x_1,\ldots,x_b\}}} \mathrm{T}_\gamma(x, L(D' \cup \{x_1',\ldots,x_b'\}))
$$

$$
+ \mathop{\mathbb{E}}_{\substack{x_1,\ldots,x_b \sim \mathcal{N}_r(x) \cap D \\ x_1',\ldots,x_b' \sim \overline{\mathcal{N}}_r(x) \\ D'=D\setminus\{x_1,\ldots,x_b\}}} \delta^{(x,D')}_{(x_1',\ldots,x_b')}
$$

Simplifying:

$$
\min_{\substack{D' \sim \mathcal{B}_b(D,\mathcal{N}_r(x)) \\ \theta=\mathcal{L}(D')}} \mathrm{T}_\gamma(x,\theta) \leq \mathop{\mathbb{E}}_{\substack{D' \sim \overline{\mathcal{B}}_b(D,\mathcal{N}_r(x),\overline{\mathcal{N}}_r(x)) \\ \theta=\mathcal{L}(D')}} \mathrm{T}_\gamma(x,\theta)) + \mathop{\mathbb{E}}_{\substack{x_1,\ldots x_b \sim \mathcal{N}_r(x) \cap D \\ x_1',\ldots,x_b' \sim \overline{\mathcal{N}}_r(x) \\ D'=D\setminus\{x_1,\ldots,x_b\}}} \delta^{(x,D')}_{(x_1',\ldots,x_b')}
$$

$\square$

**Lemma A.5.** *Lower bound on MI score under non-neighbor poisons.*

$$
\max_{\substack{D' \sim \mathcal{B}_b(D,\overline{\mathcal{N}}_r(x)) \\ \theta=\mathcal{L}(D')}} \mathrm{T}_\gamma(x,\theta) \geq \mathop{\mathbb{E}}_{\substack{D' \sim \overline{\mathcal{B}}_b(D,\overline{\mathcal{N}}_r(x),\mathcal{N}_r(x)) \\ \theta=\mathcal{L}(D')}} \mathrm{T}_\gamma(x,\theta)) - \mathop{\mathbb{E}}_{\substack{x_1,\ldots x_b \sim \overline{\mathcal{N}}_r(x) \\ x_1',\ldots,x_b' \sim \mathcal{N}_r(x) \\ D'=D\setminus\{x_1,\ldots,x_b\}}} \delta^{(x,D')}_{(x_1',\ldots,x_b')}.
$$

*Proof.* For $x_1',\ldots,x_b' \sim \mathcal{N}_r(x)$, we know,

$$
|\mathrm{T}_\gamma(x, L(D \cup \{x_1',\ldots,x_b'\})) - \mathrm{T}_\gamma(x, L(D \cup \{\texttt{PoisonMap}^b_{(x,D)}((x_1',\ldots,x_b'),\overline{\mathcal{N}}_r(x))\})| = \delta^{(x,D)}_{(x_1',\ldots,x_b')}
$$

Now, to extend this for dataset substitutions, for $x_1,\ldots,x_b \sim \overline{\mathcal{N}}_r(x) \cap D$ and $D' = D \setminus \{x_1,\ldots x_b\}$, we have

$$
|\mathrm{T}_\gamma(x, L(D' \cup \{x_1',\ldots,x_b'\})) - \mathrm{T}_\gamma(x, L(D' \cup \{\texttt{PoisonMap}^b_{(x,D')}((x_1',\ldots,x_b'),\overline{\mathcal{N}}_r(x))\})| = \delta^{(x,D')}_{(x_1',\ldots,x_b')}
$$

Taking expectation over points, we get

$$
\mathop{\mathbb{E}}_{\substack{x_1,\ldots,x_b \sim \overline{\mathcal{N}}_r(x) \cap D \\ x_1',\ldots,x_b' \sim \mathcal{N}_r(x) \\ D'=D\setminus\{x_1,\ldots,x_b\}}} \Big| \mathrm{T}_\gamma(x, L(D' \cup \{x_1',\ldots,x_b'\}))
$$

$$
- \mathrm{T}_\gamma(x, L(D' \cup \{\texttt{PoisonMap}^b_{(x,D')}((x_1',\ldots,x_b'),\overline{\mathcal{N}}_r(x))\})) \Big|
$$

$$
= \mathop{\mathbb{E}}_{\substack{x_1,\ldots,x_b \sim \overline{\mathcal{N}}_r(x) \cap D \\ x_1',\ldots,x_b' \sim \mathcal{N}_r(x) \\ D'=D\setminus\{x_1,\ldots,x_b\}}} \delta^{(x,D')}_{(x_1',\ldots,x_b')}
$$

Using Jensen's Inequality:

$$\left| \mathop{\mathbb{E}}_{\substack{x_1,\ldots,x_b\sim\overline{\mathcal{N}}_r(x)\cap D\\ x'_1,\ldots,x'_b\sim\mathcal{N}_r(x)\\ D'=D\setminus\{x_1,\ldots,x_b\}}} \mathtt{T}_\gamma(x, L(D'\cup\{x'_1,\ldots,x'_b\}))\right.$$

$$\left.- \mathop{\mathbb{E}}_{\substack{x_1,\ldots,x_b\sim\overline{\mathcal{N}}_r(x)\cap D\\ x'_1,\ldots,x'_b\sim\mathcal{N}_r(x)\\ D'=D\setminus\{x_1,\ldots,x_b\}}} \mathtt{T}_\gamma\big(x, L(D'\cup\{\mathtt{PoisonMap}^b_{(x,D')}((x'_1,\ldots,x'_b),\overline{\mathcal{N}}_r(x))\})\big)\right|$$

$$\leq \mathop{\mathbb{E}}_{\substack{x_1,\ldots,x_b\sim\overline{\mathcal{N}}_r(x)\cap D\\ x'_1,\ldots,x'_b\sim\mathcal{N}_r(x)\\ D'=D\setminus\{x_1,\ldots,x_b\}}} \delta^{(x,D')}_{(x'_1,\ldots,x'_b)}$$

Taking one side of the absolute value:

$$\mathop{\mathbb{E}}_{\substack{x_1,\ldots,x_b\sim\overline{\mathcal{N}}_r(x)\cap D\\ x'_1,\ldots,x'_b\sim\mathcal{N}_r(x)\\ D'=D\setminus\{x_1,\ldots,x_b\}}} \mathtt{T}_\gamma\big(x, L(D'\cup\{\mathtt{PoisonMap}^b_{(x,D')}((x'_1,\ldots,x'_b),\overline{\mathcal{N}}_r(x))\})\big)$$

$$\geq \mathop{\mathbb{E}}_{\substack{x_1,\ldots,x_b\sim\overline{\mathcal{N}}_r(x)\cap D\\ x'_1,\ldots,x'_b\sim\mathcal{N}_r(x)\\ D'=D\setminus\{x_1,\ldots,x_b\}}} \mathtt{T}_\gamma\big(x, L(D'\cup\{x'_1,\ldots,x'_b\})\big)$$

$$- \mathop{\mathbb{E}}_{\substack{x_1,\ldots,x_b\sim\overline{\mathcal{N}}_r(x)\cap D\\ x'_1,\ldots,x'_b\sim\mathcal{N}_r(x)\\ D'=D\setminus\{x_1,\ldots,x_b\}}} \delta^{(x,D')}_{(x'_1,\ldots,x'_b)}$$

Then,

$$\mathop{\max}_{\substack{x_1,\ldots,x_b\sim\overline{\mathcal{N}}_r(x)\cap D\\ x'_1,\ldots,x'_b\sim\overline{\mathcal{N}}_r(x)\\ D'=D\setminus\{x_1,\ldots,x_b\}}} \mathtt{T}_\gamma(x, L(D'\cup\{x'_1,\ldots,x'_b\}))$$

$$\geq \mathop{\mathbb{E}}_{\substack{x_1,\ldots,x_b\sim\overline{\mathcal{N}}_r(x)\cap D\\ x'_1,\ldots,x'_b\sim\mathcal{N}_r(x)\\ D'=D\setminus\{x_1,\ldots,x_b\}}} \mathtt{T}_\gamma\big(x, L(D'\cup\{x'_1,\ldots,x'_b\})\big)$$

$$- \mathop{\mathbb{E}}_{\substack{x_1,\ldots,x_b\sim\overline{\mathcal{N}}_r(x)\cap D\\ x'_1,\ldots,x'_b\sim\mathcal{N}_r(x)\\ D'=D\setminus\{x_1,\ldots,x_b\}}} \delta^{(x,D')}_{(x'_1,\ldots,x'_b)}$$

Simplifying:

$$\mathop{\max}_{\substack{D'\sim\mathcal{B}_b(D,\overline{\mathcal{N}}_r(x))\\ \theta=\mathcal{L}(D')}} \mathtt{T}_\gamma(x,\theta) \geq \mathop{\mathbb{E}}_{\substack{D'\sim\overline{\mathcal{B}}_b(D,\overline{\mathcal{N}}_r(x),\mathcal{N}_r(x))\\ \theta=\mathcal{L}(D')}} \mathtt{T}_\gamma(x,\theta)) - \mathop{\mathbb{E}}_{\substack{x_1,\ldots x_b\sim\overline{\mathcal{N}}_r(x)\cap D\\ x'_1,\ldots,x'_b\sim\mathcal{N}_r(x)\\ D'=D\setminus\{x_1,\ldots,x_b\}}} \delta^{(x,D')}_{(x'_1,\ldots,x'_b)}$$

$\square$

**Lemma A.6.** *Advantage of MI under poisoning.*

$$\int\limits_{\gamma=0}^{\infty} \left( Sensitivity_{b_1}\left(T_\gamma, x\right) d\gamma + Specificity_{b_2}\left(T_\gamma, x\right) d\gamma - 1 \right)$$

$$\leq \mathop{\mathbb{E}}_{\substack{D\sim\mathcal{D} \\ D'\sim\tilde{\mathcal{B}}_{b_1}(D,\mathcal{N}_r(x),\overline{\mathcal{N}}_r(x)) \\ \theta=\mathcal{L}(D')}} \left(T_\gamma(x,\theta) \mid x \in_{\mathcal{N}_r} D\right) - \mathop{\mathbb{E}}_{\substack{D\sim\mathcal{D} \\ D'\sim\tilde{\mathcal{B}}_{b_2}(D,\overline{\mathcal{N}}_r(x),\mathcal{N}_r(x)) \\ \theta=\mathcal{L}(D')}} \left(T_\gamma(x,\theta) \mid x \notin_{\mathcal{N}_r} D\right)$$

$$+ \mathop{\mathbb{E}}_{\substack{x_1,\dots,x_{b_1}\sim\mathcal{N}_r(x) \\ x'_1,\dots,x'_{b_1}\sim\overline{\mathcal{N}}_r(x) \\ D'=D\setminus\{x_1,\dots,x_{b_1}\}}} \delta^{(x,D')}_{(x'_1,\dots,x'_{b_1})} + \mathop{\mathbb{E}}_{\substack{x_1,\dots,x_{b_2}\sim\overline{\mathcal{N}}_r(x) \\ x'_1,\dots,x'_{b_2}\sim\mathcal{N}_r(x) \\ D'=D\setminus\{x_1,\dots,x_{b_2}\}}} \delta^{(x,D')}_{(x'_1,\dots,x'_{b_2})}$$

*Proof.* Using Lemma A.1, we get

$$\mathop{\mathbb{E}}_{D\sim\mathcal{D}} \left( \mathop{\min}_{\substack{D'\sim\mathcal{B}_{b_1}(D,\mathcal{N}_r(x)) \\ \theta=\mathcal{L}(D')}} T_\gamma(x,\theta) \;\middle|\; x \in_{\mathcal{N}_r} D \right)$$

$$= \int_{\gamma=0}^{\infty} P_{D\sim\mathcal{D}} \left( \mathop{\min}_{\substack{D'\sim\mathcal{B}_{b_1}(D,\mathcal{N}_r(x)) \\ \theta=\mathcal{L}(D')}} T_\gamma(x,\theta) > \gamma \;\middle|\; x \in_{\mathcal{N}_r} D \right) d\gamma$$

$$= \int_{\gamma=0}^{\infty} \mathrm{Sens}_{b_1}\left(T_\gamma, x\right) d\gamma$$

Similarly,

$$\mathop{\mathbb{E}}_{D\sim\mathcal{D}} \left( \mathop{\max}_{\substack{D'\sim\mathcal{B}_{b_2}(D,\overline{\mathcal{N}}_r(x)) \\ \theta=\mathcal{L}(D')}} T_\gamma(x,\theta) \;\middle|\; x \notin_{\mathcal{N}_r} D \right)$$

$$= \int_{\gamma=0}^{\infty} P_{D\sim\mathcal{D}} \left( \mathop{\max}_{\substack{D'\sim\mathcal{B}_{b_2}(D,\overline{\mathcal{N}}_r(x)) \\ \theta=\mathcal{L}(D')}} T_\gamma(x,\theta) \leq \gamma \;\middle|\; x \notin_{\mathcal{N}_r} D \right) d\gamma - 1$$

$$= \int_{\gamma=0}^{\infty} \left(1 - \mathrm{Spec}_{b_2}\left(T_\gamma, x\right)\right) d\gamma$$

Now, using Lemma A.4 and Lemma A.5,

$$\int_{\gamma=0}^{\infty} \left( \mathrm{Sens}_{b_1}\left(T_\gamma, x\right) + \mathrm{Spec}_{b_2}\left(T_\gamma, x\right) - 1 \right) d\gamma$$

$$= \mathop{\mathbb{E}}_{D\sim\mathcal{D}} \left( \mathop{\min}_{\substack{D'\sim\mathcal{B}_{b_1}(D,\mathcal{N}_r(x)) \\ \theta=\mathcal{L}(D')}} T_\gamma(x,\theta) \;\middle|\; x \in_{\mathcal{N}_r} D \right) - \mathop{\mathbb{E}}_{D\sim\mathcal{D}} \left( \mathop{\max}_{\substack{D'\sim\mathcal{B}_{b_2}(D,\overline{\mathcal{N}}_r(x)) \\ \theta=\mathcal{L}(D')}} T_\gamma(x,\theta) \;\middle|\; x \notin_{\mathcal{N}_r} D \right)$$

$$\leq \mathop{\mathbb{E}}_{\substack{D\sim\mathcal{D} \\ D'\sim\tilde{\mathcal{B}}_{b_1}(D,\mathcal{N}_r(x),\overline{\mathcal{N}}_r(x)) \\ \theta=\mathcal{L}(D')}} \left(T_\gamma(x,\theta) \mid x \in_{\mathcal{N}_r} D\right) - \mathop{\mathbb{E}}_{\substack{D\sim\mathcal{D} \\ D'\sim\tilde{\mathcal{B}}_{b_2}(D,\overline{\mathcal{N}}_r(x),\mathcal{N}_r(x)) \\ \theta=\mathcal{L}(D')}} \left(T_\gamma(x,\theta) \mid x \notin_{\mathcal{N}_r} D\right)$$

$$+ \mathop{\mathbb{E}}_{\substack{x_1,\dots,x_{b_1}\sim\mathcal{N}_r(x) \\ x'_1,\dots,x'_{b_1}\sim\overline{\mathcal{N}}_r(x) \\ D'=D\setminus\{x_1,\dots,x_{b_1}\}}} \delta^{(x,D')}_{(x'_1,\dots,x'_{b_1})} + \mathop{\mathbb{E}}_{\substack{x_1,\dots,x_{b_2}\sim\overline{\mathcal{N}}_r(x) \\ x'_1,\dots,x'_{b_2}\sim\mathcal{N}_r(x) \\ D'=D\setminus\{x_1,\dots,x_{b_2}\}}} \delta^{(x,D')}_{(x'_1,\dots,x'_{b_2})}$$

$$\square$$

**Theorem A.7.** *(Restatement of Theorem 5.1). For a point $x$, the advantages of a test $T_\gamma$ with and without poisoning are at odds with each other, as given by:*

$$\int\limits_{\gamma=0}^{\infty}\big(Sens_{b_1}(T_\gamma,x)+Spec_{b_2}(T_\gamma,x)-1\big)d\gamma+\int\limits_{\gamma=0}^{\infty}\big(Sens(T_\gamma,x)+Spec(T_\gamma,x)-1\big)d\gamma\leq\delta^*,$$

*where,* $\delta^*=\mathop{\mathbb{E}}\limits_{\substack{x_1,\dots x_{b_1}\sim\mathcal{N}_r(x)\\x_1',\dots,x_{b_1}'\sim\overline{\mathcal{N}}_r(x)\\D'=D\backslash\{x_1,\dots,x_{b_1}\}\\b_1=|D\cap\mathcal{N}_r(x)|}}\delta^{(x,D')}_{(x_1',\dots,x_{b_1}')}+\mathop{\mathbb{E}}\limits_{\substack{x_1,\dots x_{b_2}\sim\overline{\mathcal{N}}_r(x)\\x_1',\dots,x_{b_2}'\sim\mathcal{N}_r(x)\\D'=D\backslash\{x_1,\dots,x_{b_2}\}\\b_2=|v\cap\mathcal{N}_r(x_t)|,v\sim\mathcal{D}}}\delta^{(x,D')}_{(x_1',\dots,x_{b_2}')}.$

*Proof.* For $b_1=|\mathcal{N}_r(x)\cap D|$, we get,

$$\mathop{\mathbb{E}}\limits_{\substack{D\sim\mathcal{D}\\D'\sim\tilde{\mathcal{B}}_{b_1}(D,\mathcal{N}_r(x),\overline{\mathcal{N}}_r(x))\\\theta=\mathcal{L}(D')}}\big(\mathrm{T}_\gamma(x,\theta)\mid x\in_{\mathcal{N}_r}D\big)=\mathop{\mathbb{E}}\limits_{\substack{D\sim\mathcal{D}\\\theta=\mathcal{L}(D)}}\big(\mathrm{T}_\gamma(x,\theta)\mid x\notin_{\mathcal{N}_r}D\big)$$

Similarly, for $b_2=|v\cap\mathcal{N}_r(x_t)|,v\sim\mathcal{D}$, we get,

$$\mathop{\mathbb{E}}\limits_{\substack{D\sim\mathcal{D}\\D'\sim\tilde{\mathcal{B}}_{b_2}(D,\overline{\mathcal{N}}_r(x),\mathcal{N}_r(x))\\\theta=\mathcal{L}(D')}}\big(\mathrm{T}_\gamma(x,\theta)\mid x\notin_{\mathcal{N}_r}D\big)=\mathop{\mathbb{E}}\limits_{\substack{D\sim\mathcal{D}\\\theta=\mathcal{L}(D)}}\big(\mathrm{T}_\gamma(x,\theta)\mid x\in_{\mathcal{N}_r}D\big)$$

Applying this to Lemma A.6, we get the result. $\square$

# B  Additional Results

Table 6: Natural and robust TPR @1% FPR of MI tests on the AI4Privacy and AGNews canary datasets.

| $\mathcal{N}_r$ | MI Test | AI4Privacy | | | | | AGNews | | | | |
|---|---|---|---|---|---|---|---|---|---|---|---|
| | | Natural | Token Dropouts | Casing Flips | Chunk | PoisonM | Natural | Token Dropouts | Casing Flips | Chunk | PoisonM |
| 7-gram | LOSS | 0.104 | 0.044 | 0.046 | 0.093 | **0.004** | 0.069 | 0.012 | 0.018 | 0.085 | **0.000** |
| | kmin | 0.089 | 0.085 | 0.087 | 0.083 | **0.065** | 0.007 | 0.014 | 0.016 | **0.007** | 0.014 |
| | zlib | 0.097 | 0.061 | 0.080 | 0.099 | **0.034** | 0.034 | 0.021 | 0.027 | 0.039 | **0.005** |
| | perturb | 0.104 | 0.040 | 0.061 | 0.089 | **0.004** | 0.041 | 0.009 | 0.021 | 0.067 | **0.000** |
| | reference | 0.108 | 0.046 | 0.051 | 0.123 | **0.004** | 0.069 | 0.009 | 0.021 | 0.087 | **0.000** |
| Exact Match | LOSS | 0.030 | **0.000** | **0.000** | **0.000** | **0.000** | 0.048 | **0.000** | **0.000** | **0.000** | **0.000** |
| | kmin | 0.008 | 0.008 | 0.008 | 0.004 | **0.002** | 0.006 | 0.006 | 0.006 | 0.004 | **0.002** |
| | zlib | 0.024 | **0.000** | **0.000** | **0.000** | **0.000** | 0.014 | **0.000** | **0.000** | **0.000** | **0.000** |
| | perturb | 0.048 | **0.000** | **0.000** | **0.000** | **0.000** | 0.046 | **0.000** | **0.000** | **0.000** | **0.000** |
| | reference | 0.062 | **0.000** | **0.000** | **0.000** | **0.000** | 0.066 | **0.000** | **0.000** | **0.000** | **0.000** |
| Embedding | LOSS | 0.039 | 0.031 | 0.047 | 0.035 | **0.022** | 0.036 | 0.008 | 0.024 | 0.006 | **0.005** |
| | kmin | 0.038 | 0.036 | 0.038 | **0.014** | 0.042 | 0.010 | **0.002** | 0.008 | 0.005 | **0.002** |
| | zlib | 0.042 | 0.042 | 0.047 | 0.033 | **0.022** | 0.014 | 0.010 | 0.011 | 0.008 | **0.003** |
| | perturb | 0.069 | 0.038 | 0.071 | 0.053 | **0.016** | 0.038 | 0.016 | 0.024 | **0.003** | 0.006 |
| | reference | 0.080 | 0.025 | 0.097 | 0.042 | **0.017** | 0.027 | 0.008 | 0.024 | **0.003** | 0.005 |
| Edit Distance | LOSS | 0.046 | 0.046 | 0.037 | 0.029 | **0.027** | 0.058 | 0.054 | 0.050 | 0.023 | **0.006** |
| | kmin | 0.050 | 0.039 | **0.029** | 0.044 | 0.031 | 0.006 | 0.006 | 0.008 | 0.006 | **0.004** |
| | zlib | 0.033 | 0.039 | 0.031 | 0.023 | **0.015** | 0.037 | 0.033 | 0.033 | 0.029 | **0.006** |
| | perturb | 0.058 | 0.064 | 0.052 | 0.058 | **0.017** | 0.045 | 0.072 | 0.062 | 0.023 | **0.004** |
| | reference | 0.087 | 0.054 | 0.046 | 0.021 | **0.000** | 0.068 | 0.050 | 0.054 | 0.019 | **0.000** |

Table 7: Natural and robust membership inference test AUC scores using $n$-gram ($k$=7) neighborhood definition across different Pythia model sizes for 2.7B / 6.9B / 12B parameters on AI4Privacy.

| MI Test | Natural | PoisonM |
|---|---|---|
| LOSS | 0.568 / 0.587 / 0.587 | 0.353 / 0.252 / 0.257 |
| kmin | 0.560 / 0.561 / 0.563 | 0.472 / 0.408 / 0.423 |
| zlib | 0.554 / 0.564 / 0.564 | 0.443 / 0.375 / 0.379 |
| perturb | 0.583 / 0.600 / 0.603 | 0.380 / 0.274 / 0.286 |
| reference | 0.624 / 0.647 / 0.645 | 0.175 / 0.089 / 0.089 |

Table 8: Natural and robust AUC scores for MI tests using the $n$-gram ($k$=7) neighborhood on AI4Privacy with a larger background dataset of 1M WikiText points.

| MI Test | Natural | PoisonM |
|---|---|---|
| LOSS | 0.568 | 0.396 |
| kmin | 0.556 | 0.478 |
| zlib | 0.552 | 0.467 |
| perturb | 0.572 | 0.401 |
| reference | 0.604 | 0.232 |

## C  Additional Details About `PoisonM`

**Neighborhoods.**  We consider the following popular neighborhood definitions:

`N-gram(n=k):`   For a given sequence of $x = x_1, \cdots, x_n$, let $n\text{-gram}(x,k) = \{x_i : x_{i+k}\}_{i=1}^{n-k}$ denote the set of all $k$-grams of $x$. Then, the $n$-gram neighborhood yields the set of all sequences that share an $k$-gram with $x$: $\{x' \in \mathcal{X} \mid n\text{-gram}(x',k) \cap n\text{-gram}(x,k) \neq \emptyset\}$.
`Embedding(cosine_sim=c):`   Let $E : \mathcal{X} \to \mathbb{R}^d$ denote an embedding function that maps sequences to $d$-dimensional representations that capture their "semantics". Then, for a given sequence of $x$, the embedding similarity neighborhood yields the set of all sequences with embeddings of cosine similarity at least $c$ to $x$: $\{x' \in \mathcal{X} \mid \frac{E(x') \cdot E(x)}{||E(x)|| \, ||E(x')||} \geq c\}$.
`ExactMatch:`   For a given sequence $x$, the exact matching neighborhood yields the singleton comprising the sequence itself, i.e., $\{x\}$.
`EditDistance(distance=l):`   For a given sequence $x$, the edit distance neighborhood yields the set of all sequences that are within a normalized Levenshtein distance $l$ from $x$: $\{x' \in \mathcal{X} \mid \frac{\texttt{lev}(x,x')}{|x|+|x'|} \leq l\}$.

**Finding Poison Non-Neighbors.** We sample an actual neighbor as $x_t$ itself, and then:
1.  `N-gram(n=k):`   Iteratively (a) select a token uniformly at random, (b) replace it with a token from the vocabulary such that last-layer activations of resulting sequence have maximum cosine similarity with activations of $x_t$, where activations are computed using model that will be trained on the poison, and (c) repeat until $n$-gram overlap between the current poison and $x_t$ is less than $k$. Here we can set $\lambda = 0$ since replacing tokens automatically breaks up $n$-grams for free.
2.  `Embedding(cosine_sim=c):`   Iteratively (a) select a token uniformly at random, (b) pick the token that both maximizes activation cosine similarity and also minimizes (weighted by factor of $\lambda = -1.5$) cosine similarity of the embedding (from $E$) of the resulting sequence with that of $x_t$.
3.  `EditDistance(distance=l):`   Same procedure as that for embeddings, except we now maximize the edit distance ($\lambda = 1.5$) instead of minimizing embedding cosine similarities.
4.  `ExactMatch:`   Same procedure as that for $n$-grams, but only a single iteration.

**Finding Poison Neighbors.** We sample an actual non-neighbor as a random text from any auxiliary dataset, and then:
1.  `N-gram(n=k):`   Inject a $k$-gram from $x$ at random index (shortest $k$-gram in characters).
2.  `Embedding(cosine_sim=c):`   Iteratively (a) select a token uniformly at random, and (b) replace it with a token from vocabulary that maximizes cosine similarity of the embedding (under $E$) of the resulting sequence with $x_t$'s embedding. (c) repeat until cosine similarity exceeds $c$.
3.  `EditDistance(distance=l):`   Iteratively (a) randomly insert, delete, or substitute characters (b) greedily keep the mutation only if it decreases edit distance. (c) repeat until edit distance drops below $l$.
4.  `ExactMatch:`   Worst-case neighbors do not exist under exact matching, since the neighborhood ball holds a radius of 0.

## D  Societal Impacts

This work presents a novel poisoning vulnerability that could be used by a real-world adversary to manipulate the outcome of a high-stakes membership inference test. This could have legal and reputation-related implications. However, we believe it is important to release our findings so that (a) auditors and other entities that may wish to use membership testing may be made aware of its pitfalls, (b) the community can work towards better defining membership.

## E  Compute

We run all experiments on a machine with 4 NVIDIA H100 GPUs, 40 Intel(R) Xeon(R) Silver 4410T CPUs, and 126GB of RAM. Finetuning models typically took 2 hours, and generating poisoned datasets took between a few minutes to at most 2 hours, depending on choice of neighborhood.

