# OpenReview forum: "What Really is a Member? Discrediting Membership Inference via Poisoning"
_NeurIPS.cc/2025/Conference — NeurIPS 2025 poster_

### Official Review · Reviewer_Zgmt · 2025-07-01

**Clarity:** 3
**Significance:** 3
**Originality:** 4
**Rating:** 4
**Confidence:** 1

**Summary:**

This paper investigates the vulnerability of membership inference (MI) tests used to detect whether data points were part of a machine learning model's training set. The study demonstrates that MI tests remain susceptible to manipulation even when enhanced with relaxed membership definitions that consider approximate matches rather than exact data point correspondences.
The work presents PoisonM, a data poisoning methodology that strategically modifies training datasets to induce misclassification by MI tests, while preserving the ground-truth membership status of target data points.
The research contributions include:
– Establishing formal robustness definitions for MI tests against adversarial training set perturbations.
– Developing practical algorithms for constructing poisoned training datasets that compromise MI test effectiveness.
– Demonstrating a fundamental accuracy-robustness trade-off, where higher MI test performance on clean data correlates with increased vulnerability to adversarial manipulation.
– Providing comprehensive experimental validation showing MI test degradation under PoisonM across diverse model architectures and evaluation protocols.

**Questions:**

1) How the choice of neighborhood definition (n-gram overlap vs. embedding similarity ) influence the performance of PoisonM?

2) How feasible is it for an adversary to insert poisoned examples into public data sources in a way that ensures they are included during large-scale model training?

**Ethical Concerns:**

["NO or VERY MINOR ethics concerns only"]

**Limitations:**

yes

**Paper Formatting Concerns:**

no formatting concerns were found

**Quality:**

4

**Strengths And Weaknesses:**

Strengths

Quality
Theoretical Rigor: The paper clearly defines core concepts like neighborhood-based membership, robustness to changes in the training data, and mapping error. It builds a solid mathematical foundation for the attack and introduces pointwise robustness and Theorem 5.1 to support its claims. Also I have to add that most of the math in this paper was beyond me and could not follow it closely.

Comprehensive Experimental Analysis: The theoretical contributions are substantiated through extensive empirical evaluation, examining PoisonM effectiveness across diverse MI attack variants, benchmark datasets, model architectures, and similarity metric configurations. The experimental findings demonstrate persistent vulnerability in MI attacks despite relaxed membership criterion implementations.

General-Purpose Attack: PoisonM is flexible and works across various MI tests and neighborhood definitions, making it a broadly applicable and reproducible attack strategy.

Clarity
Clear Motivation: The connection to real-world applications (e.g., copyright enforcement, regulatory compliance) makes the work relevant and compelling. The threat model involving an adversary, challenger, and arbiter maps well to practical use cases. Structured Presentation: The paper is well-organized with clearly stated motivation, formal definitions, theoretical insights, algorithmic implementation, and experimental results.


Significance

Currently very relevant: Figuring out whether a piece of data was used to train an AI model is an important issue in current discussions about privacy and how transparent large language models (LLMs) should be. This paper shows that the tools used to answer that question can be fooled.

Policy and Legal Impact: The results raise concerns about using MI tests in real-world settings, like court cases or government audits, where accuracy really matters.

Broad Applicability: PoisonM applies to both exact and semantic definitions of membership and can be extended to other forms of training-data attribution, making it relevant to future privacy research.


Weakness:
Clarity
Dense Notation in Theoretical Sections: Definitions and theorems (particularly Theorem 5.1) use heavy notation with nested expectations and subscripts, which may be inaccessible to a broad audience.

Significance

Limited Comparisons: The experiments are strong, but the paper would be more convincing if it compared PoisonM to existing defenses or other robust MI test methods to show how well it performs in comparison.

Lack of Real-World Impact Analysis: Although the paper has clear practical relevance (like in legal or audit situations), it doesn’t show how badly MI tests might fail in real life—for example, how often they might give wrong answers or how much trust in them could be affected.

---

> ### Author Rebuttal · Authors · 2025-07-31
>
> We thank the reviewer for their comments and suggestions for the paper, and provide a detailed discussion below. We appreciate that the reviewer finds the theoretical and experimental analysis to be thorough, and the motivation and significance to be compelling.
>
> >Dense Notation in Theoretical Sections: Definitions and theorems (particularly Theorem 5.1) use heavy notation with nested expectations and subscripts, which may be inaccessible to a broad audience.
>
> We will work on improving the style and clarity of the theorem to help with readability.
>
> >The experiments are strong, but the paper would be more convincing if it compared PoisonM to existing defenses or other robust MI test methods to show how well it performs in comparison.
>
> To the best of our knowledge, we are the first to introduce a new threat model for undermining MI tests—*via data poisoning*. This is a previously unexplored attack vector, and per our knowledge there is no defense yet. Furthermore, our attack is MI-test agnostic - i.e., for a given definition of neighborhood, the same poison works for *all* MI tests for a target point(s). This is validated by our experiments as well - the same poison is effective in fooling all 5 SoTA MI attacks.
>
> Nevertheless, we explore a potential defense strategy and include additional experimental results. The intuition is that the poisoned data points might appear unnatural compared to real data. One natural strategy is to filter out suspicious points using a threshold based on the perplexity score. However, PoisonM can incorporate such defenses directly into its loss function, making it a strong,  general-purpose attack and hard to filter out. To demonstrate this,  we simulate a defense that filters out points based on their perplexity score (threshold selected to avoid false positives; 99th percentile), and show that PoisonM still successfully fools MI tests under this constraint.
>
> |  | Natural | PoisonM |
> | --- | --- | --- |
> | LOSS | 0.587 | 0.322 |
> | KMin | 0.561  | 0.444 |
> | Zlib | 0.564  | 0.420 |
> | Perturb | 0.600 | 0.340 |
> | Ref | 0.647  | 0.143 |
>
> >How the choice of neighborhood definition (n-gram overlap vs. embedding similarity ) influence the performance of PoisonM?]
>
> The choice of neighborhood function directly impacts the misalignment between the neighborhood-based definition of *true* membership and the actual influence region that determines the MI test’s behavior. Since the score of MI tests are typically computed using the model’s internal representation of a point, neighborhoods based on embedding similarity (which is a model’s internal representation of data), tend to outperform simpler, model-agnostic  definitions like n-gram overlap.
>
> >How feasible is it for an adversary to insert poisoned examples into public data sources in a way that ensures they are included during large-scale model training?
>
> Poisoning public data sources is not just a theoretical concern—it has already been demonstrated in practice. Carlini et al. [1] show that poisoning web-scale datasets is feasible and effective through real-world attacks. Our work assumes a similarly capable adversary who can manipulate public data sources to insert poisoned examples, making the threat both realistic and actionable. In particular, the adversary can adopt the same *frontrunning* strategy used by Carlini et al. to inject poisoned content into publicly scraped sources like Wikipedia backups.
>
> [1] Carlini, Nicholas, et al. "Poisoning web-scale training datasets is practical." *2024 IEEE Symposium on Security and Privacy (SP)*. IEEE, 2024.

---

### Official Review · Reviewer_YT5m · 2025-07-02

**Clarity:** 3
**Significance:** 2
**Originality:** 3
**Rating:** 5
**Confidence:** 3

**Summary:**

The paper aims to demonstrate the unreliability of membership inference tests. A membership inference test takes a trained model and an item of data and returns a score which indicates whether the item was used in the training process of the model. The approach taken in the paper is to demonstrate that it is possible to poison the training dataset such that an MI test gives incorrect predictions, e.g. an adversary may place text on their personal website which, when scraped, results in a model which is trained on that website returning a positive result for some training data which is not used in the training process.

The authors give a concrete example of an attack which can induce an incorrect prediction from an MI test and show that it works with leading MI tests and models.

**Questions:**

1. I am confused as the real-world applicability of the threat presented in the paper. In the example given, an adversary plants a poison such that the MI test predicts that there were Larry Lobster novels in the training corpus, triggering a lawsuit. However, couldn't the developer offer access to the dataset to a trusted third party, who could verify that the Larry Lobster novels are not in the dataset?
2. In a data poisoning setup, a standard threat model is to assume that an adversary can add data to a dataset but not change it. For example, a user could add fake news articles to their own website but could not edit news articles scraped from CNN.com. Does this present any challenges to your approach, which relies on substituting data in a dataset?
3. Why do the membership tests in figure 2 have such low AUC in general? They appear to be quite bad tests.

**Ethical Concerns:**

["NO or VERY MINOR ethics concerns only"]

**Final Justification:**

This is an interesting treatment of the problem of fooling membership inference tests. While the setting is slightly niche, the results, which are both theoretical and practical, are solid and a good contribution. I had some questions about the setting, mostly due to my own limited knowledge about the area. I'm happy with the authors' answers to the questions I had.

**Limitations:**

Yes

**Quality:**

3

**Strengths And Weaknesses:**

### Strengths
+ The paper studies a novel (to my knowledge) setting which could have much real-world significance.
+ The paper is generally clear and well-structured. The threat model is clearly laid out, even if I am not sure of the real-world significance.
+ The paper provides code which is good for reproducibility
+ The approach, PoisonM, is straightforward but clearly effective.

### Weaknesses
+ I am unsure of the real-world applicability of these neighborhood-based MI tests. I understand that there is significant previous work on the subject, but it does not seem really applicable for e.g. copyright if the training dataset contains an entry with a 7-gram overlap from a copyrighted source: the dataset could include a quote from the copyrighted source or indeed an independently written passage which shares the 7-gram overlap.
+ The paper seem a bit over-written, e.g. I'm not sure that the threat model / security game section adds very much to the paper and could be replaced with some more experimental results.

---

> ### Author Rebuttal · Authors · 2025-07-31
>
> We thank the reviewer for their comments and suggestions for the paper, and provide a detailed discussion below. We appreciate that the reviewer finds the setting to be novel, and the paper to be clear and well-structured.
>
> >I am confused as the real-world applicability of the threat presented in the paper. In the example given, an adversary plants a poison such that the MI test predicts that there were Larry Lobster novels in the training corpus, triggering a lawsuit. However, couldn't the developer offer access to the dataset to a trusted third party, who could verify that the Larry Lobster novels are not in the dataset?
>
> Unfortunately, sharing the training dataset with a third party does not resolve the threat—such a third party trusted by both the parties may not always exist, and the model owner could always lie about what was actually used. Even providing a training trajectory that ends in the published model is insufficient –  Roy Chowdhury et al. [1], demonstrated that it is possible to forge training trajectories and fool verifiers into believing a false dataset was used.
>
> Moreover, sharing the dataset itself often has significant barriers, including being unacceptable, due to confidentiality, privacy, and IP concerns. Many organizations (e.g., OpenAI) treat their training data as proprietary, and are unwilling to disclose it.
>
> As such, MI tests are the de facto standard for establishing whether a given data point was used in training. These tests are already widely accepted by both regulatory bodies, such as   NIST (the U.S. standards agency) [2], UK’s ICO [3], and commercial organizations, such as Google's adoption in its tensorFlow library.
>
> [1] Amrita Roy Chowdhury, Zhifeng Kong, Kamalika Chaudhuri, On the Reliability of Membership Inference Attacks
>
> [2] Elham Tabassi, Kevin Burns, Michael Hadjimichael, Andres Molina-Markham, and Julian Sexton. 2019. A taxonomy and terminology of adversarial machine learning. National Institute of Standards and Technology (2019).
>
> [3] Sasi Kumar Murakonda and Reza Shokri. ML privacy meter: Aiding regulatory compliance by quantifying the privacy risks of machine learning. CoRR, abs/2007.09339, 2020
>
> >In a data poisoning setup, a standard threat model is to assume that an adversary can add data to a dataset but not change it. For example, a user could add fake news articles to their own website but could not edit news articles scraped from CNN.com. Does this present any challenges to your approach, which relies on substituting data in a dataset?
>
> The replacement-based threat model is well-established in prior work, both in theory [4] and in practice [5]. Theoretically, Mahloujifar et al. [4] establish the existence of strong substitution-based poisoning attacks, showing that such attacks will succeed with sufficient budget. On the practical side, Carlini et al. [5] demonstrate the feasibility of substitution-based attacks at web scale by conducting real-world attacks. Their attack uses a frontrunning strategy, where the adversary anticipates public data sources (e.g., Wikipedia backup dumps) that will be scraped for training and replaces content in advance. This is a direct motivation for our attack – our adversary can follow this exact strategy (L170) to carry out PoisonM.
>
> [4] Mahloujifar, Saeed, Dimitrios I. Diochnos, and Mohammad Mahmoody. "The curse of concentration in robust learning: Evasion and poisoning attacks from concentration of measure." *Proceedings of the AAAI Conference on Artificial Intelligence*. Vol. 33. No. 01. 2019.
>
> [5] Carlini, Nicholas, et al. "Poisoning web-scale training datasets is practical." *2024 IEEE Symposium on Security and Privacy (SP)*. IEEE, 2024.
>
> >Why do the membership tests in figure 2 have such low AUC in general? They appear to be quite bad tests.
>
> This observation aligns with a growing body of work showing that MI tests are generally weaker for LLMs [6, 7]. One key reason, as hypothesized in prior work, is the nature of LLM training—few optimization steps over massive datasets—which makes individual memorization harder to detect. To address this, Maini et al. [7] propose *dataset inference* as a stronger alternative, which aggregates MI signals across multiple points to make accurate membership predictions at the dataset-level. Notably, our attack is powerful enough to *flip even these aggregated predictions*, demonstrating its effectiveness against both pointwise and dataset-level MI tests.
>
> [6] Duan, Michael, et al. "Do Membership Inference Attacks Work on Large Language Models?." First Conference on Language Modeling.
>
> [7] Maini, Pratyush, et al. "LLM Dataset Inference: Did you train on my dataset?." *Advances in Neural Information Processing Systems* 37 (2024): 124069-124092.

---

> > ### Comment · Reviewer_YT5m · 2025-08-06
> >
> > Thanks for your reply. You've clarified the threat model, which I understand better now.
> > Although the authors have clarified that there exist threat vectors which can allow replacement-based attacks, I continue to think that their threat model would capture a larger variety of realistic cases if they considered the additive case. The replacement case is, I think, sufficient for a good contribution.

---

### Official Review · Reviewer_EmXg · 2025-07-02

**Clarity:** 2
**Significance:** 2
**Originality:** 3
**Rating:** 3
**Confidence:** 4

**Summary:**

This paper investigates the problem of membership inference tests under the neighborhood-based definition where all semantic neighbors of a training point are also treated as members. Authors proposed to study the MI test from a robust perspective through substituting neighbors or non-neighbors of target training point. And authors theoretically show that there is a trade-off between a test’s accuracy and its robustness to poisoning. Experimental results show the proposed poisoning attack can make existing MI test less effective.

**Questions:**

1. This is a metric-based membership inference, how to determine the threshold value?
2. Can authors present more clearly about worst-case datasets? And give a clear explanation about why membership inference test can make wrong predictions when the perturbed dataset is the worst-case dataset?
3. In Figure 1, what does the orange area/line mean? Is the target point a member?
4. Please give more detailed explanation about Figure 1.

**Ethical Concerns:**

["NO or VERY MINOR ethics concerns only"]

**Final Justification:**

Many thanks for the response to the concerns, which help improve the understanding. But "the poison is crafted such that any downstream MI test on the target point is fooled, regardless of its internal mechanics" still seems too strong to me, without a clear intuition. Not sure if the concept of worst-case dataset could be more formal.

**Limitations:**

yes

**Quality:**

3

**Strengths And Weaknesses:**

Strengths in this paper are summarised as follows:
1. This paper studies the problem of membership inference in a relaxed neighborhood-based assumption, i.e., all semantic neighbors of a training point can be considered as members. I believe this research perspective is more practical, especially in the ear of LLM. So, the research motivation and significance are strong.
2. Authors give thorough definitions and theoretical analysis about the problem of membership inference tests.

However, there are some weaknesses in this paper shown as follows:
1. A clear description about attacker’s background knowledge is expected. For example, does an attacker have to know the data distribution of target dataset in the scenario this paper studied? In addition, what kind of information the attacker can obtain from the target model, such that the attacker can exploit the information to implement the follow-up membership inference test?
2. The definition of 4.2 is not clear, especially the given equation of expansions of dataset.
3. A clearer description about dataset perturbation, worst-case dataset, and the reliability of membership inference test should be very important and expected. Even though authors gave clear descriptions on other definitions, explanation and justification about this part are vague.
4. The poisoning attack might only be applicable to the specific form of MI concerned in this work. Once an attacker knows how MI works (i.e., how to construct the features for member and non-member), she could alway find a way to fool MI more or less if she could manipulate/poisoning the training/tuning data.

---

> ### Author Rebuttal · Authors · 2025-07-31
>
> We thank the reviewer for their comments and suggestions for the paper, and provide a detailed discussion below. We appreciate that the reviewer finds the research motivation strong, and the theoretical analysis thorough.
>
> >A clear description about attacker’s background knowledge is expected. For example, does an attacker have to know the data distribution of target dataset in the scenario this paper studied? In addition, what kind of information the attacker can obtain from the target model, such that the attacker can exploit the information to implement the follow-up membership inference test?
>
> We emphasize that our attacker is solely responsible for poisoning the training dataset—they do not participate in or implement the membership inference (MI) test itself. The key idea is that the poison is crafted such that *any* downstream MI test on the target point is fooled, regardless of its internal mechanics. In fact, a very attractive feature of our attack is that it is MI test agnostic – i.e., the same poison works for all MI tests for the target point(s). This is validated by our experiments as well - the same poison is effective in fooling all 5 SoTA MI attacks.
>
> To construct the poison itself,  we consider a highly practical attacker that does *not* require knowledge of the underlying data distribution of the training set. It only needs to know the exact target point(s) for which they wish to flip the membership test results which is inevitable for a targeted attack. Additionally, In our experiments, since the PoisonM loss requires only the model’s last layer activations, the attacker only needs query access to the model’s logprobs which is a standard assumption in the poisoning and adversarial example literature [1]. However, even under an even more restrictive setting where query access to the target model is prohibited, an attacker can optimize against a surrogate model, with the expectation that the poison will transfer to the target model. Below, we experiment with an OLMO2-7b model trained on poisons computed against a surrogate Pythia-6.9b model.  We find that PoisonM is still effective against MI tests even in this completely blind setting, without query access to the target model.
>
> |  | Natural | PoisonM |
> | --- | --- | --- |
> | LOSS | 0.567 | 0.313 |
> | KMin | 0.542 | 0.433 |
> | Zlib | 0.554 | 0.410 |
> | Perturb | 0.587 | 0.334 |
> | Ref | 0.590 | 0.298 |
>
> [1] Wan, Alexander, et al. "Poisoning language models during instruction tuning." International Conference on Machine Learning. PMLR, 2023.
>
> >The definition of 4.2 is not clear, especially the given equation of expansions of dataset. A clearer description about dataset perturbation, worst-case dataset, and the reliability of membership inference test should be very important and expected. Even though authors gave clear descriptions on other definitions, explanation and justification about this part are vague. Can authors present more clearly about worst-case datasets? And give a clear explanation about why membership inference test can make wrong predictions when the perturbed dataset is the worst-case dataset?
>
> We will certainly add more clarifying text in the paper on these points.
>
> In Equation 4.2, given an original dataset, we define the space of (*membership-invariant) perturbations*—that is, all datasets obtained by replacing a point in the dataset with another point from the **same neighborhood class** (i.e., neighbors are replaced with neighbors, non-neighbors with non-neighbors) with respect to the target point x. These perturbations preserve the **membership label** of a target point under the chosen neighborhood definition. A simple example for the ease of illustration is as follows.  If a target sentence “John eats cake” is a member, and we use edit-distance with threshold \ell = 1, then substituting a point in the dataset with “John eatz cake” (which remains within the edit-distance based neighborhood) results in a *membership-invariant* perturbation. The target’s membership status is preserved.
>
> The **worst-case dataset** is then defined as a dataset drawn from this perturbation set but **adversarially chosen** to maximize the error of the membership inference test. In other words, while the true membership status (as defined by the neighborhood) stays the same, the attacker crafts such perturbations to **fool the MI test**—causing it to incorrectly predict the membership status for the target point. PoisonM achieves this through loss functions that optimize the representation space (e.g., embeddings) to mislead the test, even while staying within the bounds of valid dataset perturbations in the input space.
>
> >In Figure 1, what does the orange area/line mean? Is the target point a member? Please give more detailed explanation about Figure 1.
>
> In Figure 1(a), the orange area represents all points which, if included in the training set and trained upon, will cause the MI test T to predict “member” for the target point. In other words, these are the points that influence the membership prediction for the target point x under the test T. The green point represents a clean training data point which is substituted by the purple point (poisoned data point) under our attack. In Figure 1(b), the target point is not a member because the green point (and purple point) is outside the neighborhood radius. In Figure 1(c), the target point is a member, since the green point (and purple point) are inside the neighborhood radius. Overall, Figure 1 illustrates the **misalignment** between the neighborhood-based definition of membership (using any distance metric) and the **actual influence region** that determines the behavior of a membership inference test on the target point x.
>
> >This is a metric-based membership inference, how to determine the threshold value?
>
> There are two distinct thresholds relevant to our setting:
>
> - Neighborhood radius used to determine *true membership* under a given neighborhood definition. We adopt radius values consistent with those studied in prior work [2]. Specifically, we select them by calibrating on points from the canary dataset to ensure that the test performs well under natural conditions.
> - Threshold for the MI test score, which determines the test's decision boundary. In Table 2 and Figure 2, we report AUC scores to capture the MI test’s performance *across all thresholds*. Additionally, in Table 5 (Appendix B), we present results using a concrete threshold selected via the standard procedure of targeting a 1% false positive rate (FPR).
>
> [2] Duan, Michael, et al. "Do Membership Inference Attacks Work on Large Language Models?." First Conference on Language Modeling.
>
> >The poisoning attack might only be applicable to the specific form of MI concerned in this work. Once an attacker knows how MI works (i.e., how to construct the features for member and non-member), she could alway find a way to fool MI more or less if she could manipulate/poisoning the training/tuning data.
>
> We stress that an attractive feature of our attack is that it is completely MI test agnostic – i.e.,  for a given neighborhood definition (e.g., ngram), the same poison works for all MI tests for a target point(s). This is also validated by our experimental results - the same poison works against all 5 state-of-the-art MI tests.

---

> > ### Author Response · Authors · 2025-08-06
> > **Follow-up Questions**
> >
> > Dear Reviewer,
> >
> > We are happy to answer any follow-up questions and comments you might have. Thank you for your time and feedback!

---

> > ### Comment · Reviewer_EmXg · 2025-08-09
> >
> > Dear authors,
> > Many thanks for the response to the concerns, which help improve the understanding. But "the poison is crafted such that any downstream MI test on the target point is fooled, regardless of its internal mechanics" still seems too strong to me, without a clear intuition. Not sure if the concept of worst-case dataset could be more formal.

---

### Official Review · Reviewer_5Yuq · 2025-07-02

**Clarity:** 4
**Significance:** 4
**Originality:** 4
**Rating:** 5
**Confidence:** 4

**Summary:**

The authors question the validity of membership inference evaluations for LMs demonstrating that even under the relaxed neighborhood-based membership evaluations, efficient poisoning can help completely flip predictions for members and non-members alike. The authors propose concrete instantiations for poisoning schemes targeted at various downstream MIAs and support their results with theoretical analysis of the poisoning and its relationship to attack success. The authors evaluate their poisoning attack on multiple models and downstream MIAs, and even extend their poisoning attack to dataset inference

**Questions:**

- Definitions in the paper talk about "replacing" actual neighbors with other neighbors (Definition 4,2. etc) but in a realistic data poisoning setting, the adversary can only add more data, not remove/replace any data. How do these two connect? Given the way neighborhood is defined, injecting neighbors, even if they are closer to the target record, does not make existing neighbors non-neighbors. Do the authors actually replace these neighbors in their experimental setup, or is it purely for easier theoretical analysis?

- Lemma 3.5: How does this connect to/differ from the standard definition of the adversary's advantage in an cryptographic game?

- Regarding the last step of the adversarial game (under threat model): because of the additional party, A's success is now also tied to C's success for the benign case. For instance, I could not poison either data and if C's test is not strong enough (which is the case for MIAs for LLMs), A wins by default in most cases. However, theoretical analyses further down in Theorem 5.1 suggest that this advantage would be high again if the attack is too powerful. Is there then a U-shaped curve for the adversary's advantage with respect to the downstream MIA's performance?

- L230: Intuitively, this seems off. By construction, the poison data is a neighbor of X. If this poison is also indistinguishable from a clean non-member, then that clean non-member is also in the neighborhood of X, which makes it a member.

- L253: I am a little confused about the generality claims. Unless I missed it, the way I understand each poisoning experiment corresponds to adding data upon anticipating some MIA that might be used in downstream inference. Is that the case, or do the authors use one poisoning strategy and then test other MIAs on it? Some clarification would be nice. Also, why not enforce this generalization by design? The loss used for token selection could be easily adapted to consider multiple MIAs simultaneously.

- Could the authors share some of these generated poison samples to give a sense of what they look like? Given the use of token replacements I imagine them to look unnatural (high perplexity) and easily filterable with cheap alternatives, but that discussion can only continue after looking at the poison texts.

- L300: What is the level of "dataset" here for dataset inference? Please provide additional details in the main body.

**Ethical Concerns:**

["NO or VERY MINOR ethics concerns only"]

**Final Justification:**

I think with the additional experiments and clarifications, the paper will be a great addition to the conference. My recommendation remains the same (i.e., accept).

**Limitations:**

Yes, but I would encourage the authors to talk about it a bit more in the main body of the paper.

**Quality:**

4

**Strengths And Weaknesses:**

# Strengths

- I think the scenario considered here is very interesting and practical: not only can it be used by malicious data curators and model owners to bypass any membership/dataset inference tests, but also scrutinize companies and organizations by selectively poisoning Internet data with records that make models behave like they have strongly overfit on some target data.
- The theoretical analyses are explained very clearly and paired with intuition, making them easy to follow. Overall, I found the paper very well written.
- Experimental analyses are thorough, with evaluations across multiple model sizes, attacks, and even inference granularities (record and dataset level)
- I really like Figure 1! I think it gets across the point beautifully

# Weaknesses

I don't see any critical weaknesses in the paper, but there are some things that could be improved or clarified:

- The way I see it, flipping non-membership to membership is straightforward: just insert the target datapoint itself into model training data. Either the model trainer actually scans its training data to check for the record (which is unlikely but in case it does, it can share the computation as a proof of non-membership with any relevant arbiter), or it doesn't (in which case, there is no reason to be stealthy and not just insert the target record directly). It is hard to think of a practical scenario where a model trainer would want to prove that they did in fact train on some particular record in training. Making a member be classified as a non-member is particularly interesting, especially considering potentially **adversarial model trainers or data curators** that use some copyrighted data, but augment it with such poisoned data such that any downstream MI/DI evaluations make it look like such data was never used in training.

- L260: While this is understandable, I think the authors can try and at least simulate what the effect of poisoning might be by a) using a lot more data, and/or b) partitioning the data such that all poison data is injected in the first X% of fine-tuning data. Given the reasonable compute requirements (as detailed in the Appendix), it would be nice to potentially consider another open-source model family (like OLMO/OLMO2 [1]).

- The per-record effective DPR (data poisoning rate) is reasonably low: 10 poisons per record in the worst case for a 100K dataset size, i.e. 0.01% DPR. However, for a set of 500 members and non-members this number does scale up very quickly, reaching 5.5% DPR (for 500 members and 500 non-members) : quite high, especially for LLM training! This DPR would be particularly relevant for dataset inference, where the adversary would have to influence membership labels for multiple records to sway the final DI outcome. Can the authors comment on (and perhaps analyze) the tradeoff between the DPR here and final attack success, at least for the DI case (and potentially also for MI by controlling the number of poisons added for non-members)?

## Minor Comments

- L2: "language model's training set" - although this work focuses on text models, the MI definition is not specific to language models - please rephrase to clarify the distinction.
- Might be worth looking into attacks like Min-K++ [2]
- Figure 2: Please increase size and use log-log scale (focus on low-FPR region)
- L319-321: The 2.7B model has had very "unexplainable" observations in several works (acknowledged by the Pythia authors too) so I would not read too much into results for this model- maybe test if for a smaller one from the same model family?

### References
- [1] OLMo, Team, et al. "2 OLMo 2 Furious." arXiv preprint arXiv:2501.00656 (2024).
- [2] Zhang, Jingyang, et al. "Min-k%++: Improved baseline for detecting pre-training data from large language models." arXiv preprint arXiv:2404.02936 (2024).

---

> ### Author Rebuttal · Authors · 2025-07-31
>
> We thank the reviewer for their comments/suggestions, and provide discussion below. We appreciate that the reviewer finds the scenario interesting and practical, and theoretical/experimental analyses thorough.
>
> >L260: While this is understandable, I think the authors can try and at least simulate what the effect of poisoning might be by a) using a lot more .... open-source model family (like OLMO/OLMO2 [1]).
>
> Below, we run the recommended experiment of using more data with 1M points (ngram definition), and find that PoisonM continues to successfully reduce performance of the tests.
>
> |  | Natural | PoisonM |
> | --- | --- | --- |
> | LOSS | 0.568 | 0.396 |
> | KMin | 0.556 | 0.478 |
> | Zlib | 0.552 | 0.467 |
> | Perturb | 0.572 | 0.401 |
> | Ref | 0.604 | 0.232 |
>
> We also conduct experiments below on OLMO2-7b model (ngram definition), and find that here too, PoisonM successfully reduces performance of membership tests:
>
> |  | Natural | PoisonM |
> | --- | --- | --- |
> | LOSS | 0.567 | 0.276 |
> | KMin | 0.542 | 0.412 |
> | Zlib | 0.554 | 0.386 |
> | Perturb | 0.587 | 0.303 |
> | Ref | 0.590 | 0.255 |
>
> >The per-record effective DPR (data poisoning rate) is reasonably low: 10 poisons per record in the worst case .... tradeoff between the DPR here and final attack success, at least for the DI case (and potentially also for MI by controlling the number of poisons added for non-members)?
>
> We explicitly account for the tradeoff between data poisoning rate (DPR) and attack success in our formulation in Section 5—this is formally captured by **budget parameter** b, which controls number of poisons per target. Higher budget leads to lower mapping error or better attack success.
>
> As requested, we conduct additional experiments in the DI setting, focusing on low-DPR regimes. Specifically, we consider 2 reduced-DPR configurations:
>
> - 0.55% DPR: Using the setting from the previous comment’s response, with total dataset size of 1M , we achieve a **DPR of just 0.55% (for 500 members and 500 non-members)** . Even at this lower rate, DI is successfully fooled: p-values are **3e-7** for non-members and **0.99** for members.
> - 0.3% DPR: Reducing budget further to 5 poisons per record (yielding a 0.3% DPR for 500 members and 500 non-members), the attack remains effective: p-values of 2e-2 for non-members and 0.99 for members.
>
> These results highlight that even at substantially reduced poisoning rates, PoisonM can still mislead dataset inference attacks. We will add this analysis to the paper to make DPR-attack success tradeoff clearer.
>
> >Definitions in the paper talk about "replacing" actual neighbors with other neighbors (Definition 4,2. etc) but in a realistic data poisoning setting, the adversary can only add more data, not remove/replace any data. How do these two connect? Given the way neighborhood is defined, injecting neighbors, even if they are closer to the target record, does not make existing neighbors non-neighbors. Do the authors actually replace these neighbors in their experimental setup, or is it purely for easier theoretical analysis?
>
> We replace neighbors in both our experimental setup and theoretical analysis. The replacement-based threat model is well-established in prior work, both in theory [1] and in practice [2]. Theoretically, Mahloujifar et al. [1] establish the existence of strong substitution-based poisoning attacks, showing that such attacks will succeed w/ sufficient budget. On the practical side, Carlini et al. [2] demonstrate the feasibility of substitution-based attacks at web scale by conducting real-world attacks. Their attack uses a frontrunning strategy, where the adversary anticipates which public data sources (e.g., Wikipedia backup dumps) will be scraped for training and replaces content in advance. This is a direct motivation for our attack – the adversary can follow this exact strategy (L170) to carry out PoisonM.
>
> [1] Mahloujifar, Saeed, et al. The curse of concentration in robust learning: Evasion and poisoning attacks from concentration of measure. AAAI, 2019
>
> [2] Carlini, Nicholas, et al. Poisoning web-scale training datasets is practical. IEEE S&P, 2024
>
> >How does this connect to/differ from the standard definition of the adversary's advantage in an cryptographic game?
>
> Our security game is directly inspired by standard cryptographic security games and hence, their connection is natural. The adversary's advantage in a (cryptographic) distinguishing game captures their ability to differentiate between a cryptographic protocol’s output vs. random data (e.g., a good encryption scheme’s output should be indistinguishable from a random string without access to the secret key). Formally, this is quantified by benefit obtained by the adversary over simply random guessing.  We adopt the same notion of *advantage* in the context of membership inference (MI) (this was originally done by Yeom et al. [3] and many follow-up works thereafter [4,5]) – here, advantage captures tester’s gain over random guessing in determining whether a point was in the training set.
>
> [3] Yeom, Samuel, et al. "Privacy risk in machine learning: Analyzing the connection to overfitting." IEEE CSF, 2018.
>
> [4] Salem, Ahmed, et al. SoK: Let the privacy games begin! A unified treatment of data inference privacy in machine learning. IEEE S&P, 2023
>
> [5] Li, Zhuohang, et al. Analyzing inference privacy risks through gradients in machine learning. ACM CCS, 2024
>
> >Regarding the last step of the adversarial game (under threat model): because of the additional party, A's success is now also tied to C's success for the benign case. For instance, I could ..... U-shaped curve for the adversary's advantage with respect to the downstream MIA's performance?
>
> Indeed, when the test behaves poorly by itself, an adversary need not take much action to fool the test, and when the test behaves well, adversary gains significant advantage, suggesting U-shaped curve with respect to MIA’s performance. We will add discussion of this to the paper.
>
> >Intuitively, this seems off. By construction, the poison data is a neighbor of X. If this poison is also indistinguishable from a clean non-member, then that clean non-member is also in the neighborhood of X, which makes it a member.
>
> The key point is that *"indistinguishability"* here is defined **with respect to a specific MI test** T. That is, the poison is *T-equivalent* to a clean non-member, meaning that if either the poison or a clean non-member were included in the training set, the MI test T would assign (approximately) **the same score** to the target point x.
>
> In other words, while the poison is a neighbor of x in the input space as measured by the distance metric (by design), it is crafted in a way such that, from the perspective of test T, its training-time influence *mimics* that of a clean non-member. We will revise the text to clarify this.
>
> >I am a little confused about the generality claims. Unless I missed it, the way I understand each poisoning experiment corresponds to adding data upon anticipating some MIA that might be used in downstream inference. Is that the case, or do the authors use one poisoning strategy and then test other MIAs on it? Some clarification would be nice. Also, why not enforce this generalization by design? The loss used for token selection could be easily adapted to consider multiple MIAs simultaneously.
>
> For a given neighborhood definition (e.g., ngram), we use one poisoning strategy and test all MIAs on this single strategy, i.e., we do not incorporate test-specific changes to the strategy. In other words, PoisonM’s implementation is MI test agnostic. Our experimental results demonstrate that the *same* poison works against *all* 5 state-of-the-art MI tests.
>
> >Could the authors share some of these generated poison samples to give .... perplexity) and easily filterable with cheap alternatives, but that discussion can only continue after looking at the poison texts.
>
> Here are some example poisons for the AI4Privacy dataset:
>
> ”Mr. Aberspathy, I write, expressing My Interest in your forensic psychologists Research.Could you Please sharing your latest publications'? I am especially interested of Your studyabout IPV4 user 122.171,108.126.”
>
> “Hi, i am Unable to proceeding of education course payments. My MasterCard Cardnumber 0789777405804829 kept getting declined.”
>
> “Dear Eola,I've looks at your files Under Auto Loan Account and noticing some inconsistencies, Your DOB is still listing as 11/12/1977, is to correct'? Also, can You confirmed your secondary addressing atApt. 488, Yorkshire and. Humber, 81414.5215?”
>
> “Dear Ebba, we appreciate your interest in our Distance learning program. A link to, trial Course will be sent to Kristina36@ahoo.Com.”
>
> We observe that generated poison texts maintain a reasonable level of readability. Moreover, we note that PoisonM is a strong, general-purpose attack and can adapt to defenses by incorporating filtering criteria—such as perplexity filter—directly into its loss function. We demonstrate this with an additional experiment where we consider a setting in which perplexity filter is employed (threshold selected to ensure low FPR of 1%), and show PoisonM can adapt to this setting and remain effective against all considered MI tests.
>
> |  | Natural | PoisonM |
> | --- | --- | --- |
> | LOSS | 0.587 | 0.322 |
> | KMin | 0.561  | 0.444 |
> | Zlib | 0.564  | 0.420 |
> | Perturb | 0.600 | 0.340 |
> | Ref | 0.647  | 0.143 |
>
> >What is the level of "dataset" here for dataset inference?
>
> Here, we consider the setting proposed by Maini et al in the original paper [8] and use the DI procedure of aggregating MI scores from multiple points (we use 500 here) to produce a single p-value, which is then used to assign a label of {0,1} to the whole dataset (AI4Privacy or AGNews) as to whether it is in the training set or not.
>
> [6] Maini, Pratyush, et al. LLM Dataset Inference: Did you train on my dataset? NeurIPS, 2024

---

### Note · Authors · 2025-08-15

We would like to take this final opportunity to thank the reviewers and AC for the effort and time they have spent on the reviews and rebuttals. We also appreciate the recognition regarding the novelty and significance of the studied problem (**5Yuq, EmXg, YT5m, Zgmt**), the clarity of our writing (**5Yuq, YT5m, Zgmt**), and the strength of our theoretical analysis (**5Yuq, EmXg, Zgmt**) and experimental results (**5Yuq, Zgmt**).

Finally, we provide our response to the additional question posed by reviewer **EmXg**. On the discussion of providing intuition, we note that all membership tests rely upon some function of the model’s last-layer activations at/around the target point, and since our poisons are directly optimizing these representations, a single poisoning strategy (for a given neighborhood definition) suffices to achieve high attack success across all tests. This is also confirmed by our experimental results.

Best,

Authors

---

### Decision · Program_Chairs · 2025-09-17

**Decision:**

Accept (poster)

**Comment:**

The authors propose a method called PoisonM to poison a language model's training set in a way that invalidates membership inference. Given any distance metric used to define neighborhood in the LM's training data, PoisonM crafts poisoned data that, once trained on, degrades the performance of existing membership tests to below random chance.

Reviewers generally appreciated the paper's strong empirical result supported by intuitive theoretical analysis, with many practical implications for the interpretation of MIA. However, Reviewers 5Yuq and EmXg had some confusion about whether PoisonM is agnostic to the MI test or not, since an attack that requires knowledge of the MI test would significantly reduce its practicality.

During the rebuttal, the authors clarified that PoisonM only requires the attacker to choose the neighbor definition, i.e. distance metric, and crafts poisoned data based on this metric. The resulting poison is general to different MI tests, so the attacker is agnostic to the MI test itself. Given that the rebuttal sufficiently addressed this concern, AC believes the paper is ready for publication at NeurIPS and recommends acceptance.